# LIMOPro: Reasoning Refinement for Efficient and Effective Test-time Scaling

**Yang Xiao**[1]    **Jiashuo Wang**[1]    **Ruifeng Yuan**[1]    **Chunpu Xu**[1]
**Kaishuai Xu**[1]    **Wenjie Li**[1†]    **Pengfei Liu**[2,3†]
[1]The Hong Kong Polytechnic University    [2]Shanghai Jiao Tong University    [3]SII
yang-alan.xiao@connect.polyu.hk    csjwang@comp.polyu.edu.hk

## Abstract

Large language models (LLMs) have demonstrated remarkable reasoning capabilities through test-time scaling approaches, particularly when fine-tuned with chain-of-thought (CoT) data distilled from more powerful large reasoning models (LRMs). However, these reasoning chains often contain verbose elements that mirror human problem-solving, categorized as progressive reasoning (the essential solution development path) and functional elements (verification processes, alternative solution approaches, and error corrections). While progressive reasoning is crucial, the functional elements significantly increase computational demands during test-time inference. We introduce **PIR (Perplexity-based Importance Refinement)**, a principled framework that quantitatively evaluates the importance of each reasoning step based on its impact on answer prediction confidence. PIR systematically identifies and selectively prunes only low-importance functional steps while preserving progressive reasoning components, creating optimized training data that maintains the integrity of the core solution path while reducing verbosity. Models fine-tuned on PIR-optimized data exhibit superior test-time scaling properties, generating more concise reasoning chains while achieving improved accuracy (+0.9% to +6.6%) with significantly reduced token usage (-3% to -41%) across challenging reasoning benchmarks (AIME, AMC, and GPQA Diamond). Our approach demonstrates strong generalizability across different model sizes, data sources, and token budgets, offering a practical solution for deploying reasoning-capable LLMs in scenarios where efficient test-time scaling, response time, and computational efficiency are valuable constraints. Code and dataset are available at the LIMOPro.

## 1   Introduction

Large language models (LLMs) have demonstrated remarkable capabilities in complex reasoning tasks through chain-of-thought (CoT) [33], where models generate step-by-step solutions to problems. Recent advances of test-time scaling [12, 26] can significantly enhance LLMs' reasoning abilities by increasing the compute at test time. One approach to the test-time scaling involves fine-tuning LLMs on high-quality reasoning data distilled from more powerful large reasoning models (LRMs) [20, 37]. LRMs like DeepSeek-R1 [10], OpenAI o1 [12], and QwQ [29] represent the state of the art in this paradigm, producing reasoning chains that lead to accurate solutions.

However, this approach faces a significant challenge: reasoning chains distilled from LRMs often contain numerous functional elements that, while reflecting human problem-solving processes, possibly produce unnecessarily verbose outputs [7, 32, 4]. In a typical mathematical scenario, an LRM might solve a problem by establishing an initial solution path, verifying calculations, identifying

---

†Corresponding authors.

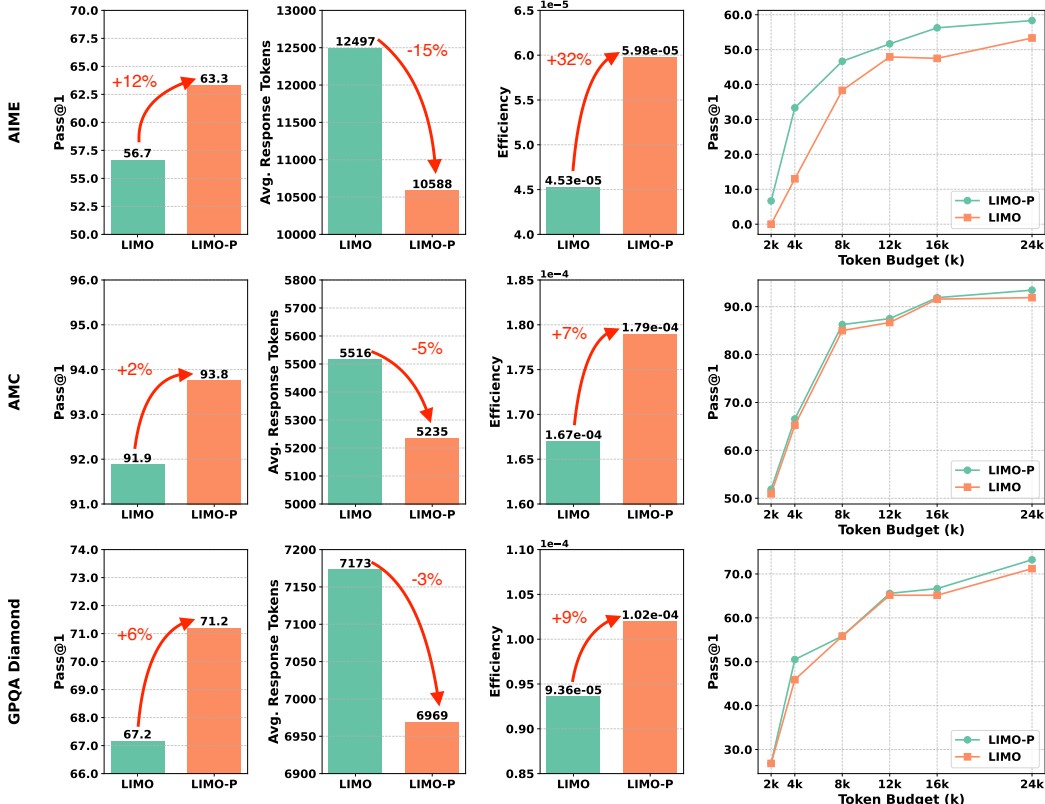

Figure 1: Our PIR framework (implemented as LIMO-P) optimizes the efficiency-effectiveness tradeoff in LLM reasoning across AIME, AMC, and GPQA Diamond benchmarks: it consistently enhances accuracy while concurrently reducing response tokens, thus improving computational efficiency, demonstrating that selectively pruning low-importance functional steps produces more concise, faster, and more accurate reasoning chains.

errors, revising the approach, and ultimately confirming the final answer. This thorough process generates lengthy reasoning chains with redundant or marginally valuable steps. When these verbose chains are used to fine-tune target models, the resulting models inevitably adopt similar behaviors, producing equally elaborate reasoning sequences despite many steps contributing minimally to solution accuracy. Consequently, inference time increases substantially, along with computational demands and response latency. This inefficiency poses a considerable obstacle to implementing reasoning-enhanced LLMs in practical applications where timely, precise responses are essential.

To address this challenge, we introduce **PIR (Perplexity-based Importance Refinement)**, a novel framework that systematically refines reasoning chains to optimize the efficiency-effectiveness balance. Our refinement approach builds upon three key innovations that work in concert: (1) We develop a systematic methodology to classify functional patterns in complex reasoning chains, identifying four distinct modes—progressive reasoning and three types of functional steps: verification, multi-method validation, and error correction. (2) Through comprehensive analysis across diverse problem domains, our analysis indicates that progressive reasoning forms the essential logical backbone, directly advancing solution derivation, and must be preserved intact. In contrast, functional steps frequently introduce computational overhead with redundancies that can be strategically pruned without compromising solution integrity. This differential treatment—preserving progressive reasoning while selectively optimizing functional elements—maintains the core problem-solving logic while significantly improving computational efficiency. (3) Building on (1) and (2), we propose the PIR metric, which quantitatively measures each functional step's contribution to the final solution by comparing answer perplexity with and without specific steps. This perplexity-based evaluation provides a principled mechanism to refine reasoning chains by identifying and selectively removing low-importance functional steps while preserving the progressive reasoning chain. By selectively targeting only non-essential functional components, our refinement approach maintains the logical coherence of the solution process while significantly reducing verbosity.

By applying our refinement framework to datasets distilled from different foundation models (LIMO [37] from Deepseek-R1, S1 [20] from Gemini Flash Thinking [2], and LIMO-V2 [37] from QwQ), we create PIR-optimized training datasets. Models fine-tuned on these refined datasets maintain or enhance accuracy while significantly reducing response length compared to models trained on the original unrefined data. Our experiments across challenging reasoning benchmarks demonstrate that PIR-refined models consistently outperform their counterparts in both effectiveness and efficiency, achieving up to 71% efficiency improvement.

The contributions of our work include: 1. A novel perplexity-based refinement framework (PIR) for quantifying the importance of reasoning steps and optimizing reasoning chains, balancing efficiency and effectiveness. 2. A systematic analysis of reasoning patterns in reasoning problem-solving, providing insights into the structure and function of different reasoning elements. 3. Comprehensive empirical validation showing that PIR-refined models achieve improved accuracy (+0.9% to +6.6%) with significantly reduced token usage (-3% to -41%) across diverse benchmarks. 4. Demonstration of the framework's generalizability across different model sizes, data sources, and token budgets.

Our work addresses a critical gap in current approaches to LLM reasoning enhancement, offering a practical solution for more efficient reasoning without sacrificing solution quality. By systematically refining training data to preserve essential reasoning while eliminating redundant functional steps (verification, validation, and error correction processes), we enable LLMs to produce concise yet equally effective reasoning chains, advancing the practical deployment of reasoning-capable LLMs across scenarios where response time and computational resources are valuable constraints.

## 2 Reasoning Refinement: Reasoning Optimization Framework

Reasoning chains produced by LRMs typically contain numerous functional steps—including verification processes, multiple solution approaches, and error corrections—that mirror human problem-solving but significantly increase computational overhead without proportionally enhancing solution accuracy. This section presents our Perplexity-based Importance Refinement (PIR) framework, which quantitatively evaluates reasoning step importance and systematically refines reasoning chains by preserving essential solution elements while removing less valuable components.

### 2.1 Problem Formulation

In this paper, we address the challenge of optimizing reasoning chains for complex reasoning tasks. Formally, we consider a dataset $\mathcal{D}$ containing question-reasoning-answer triplets $(q, r, a)$, where $q \in \mathcal{Q}$ represents a reasoning problem, $r \in \mathcal{R}$ is the reasoning chain, and $a \in \mathcal{A}$ is the answer. We define a reasoning chain $r$ as a sequence of intermediate steps $\{s_1, s_2, ..., s_n\}$, where each step $s_i$ represents a logical deduction that bridges the gap between the question and the final answer.

Our goal is to refine each reasoning chain $r$ into an optimized version $r'$ such that: (1) The answer accuracy is preserved: $f(q, r') = f(q, r) = a$; (2) The token length is reduced: $|r'| < |r|$; (3) The essential reasoning logic is maintained without harming the quality of the dataset.

### 2.2 Theoretical Foundations

**Cognitive Reasoning Patterns**    Through extensive analysis of reasoning patterns, we identify four representative distinct modes [9, 8] that characterize problem-solving processes: **(1) Progressive Reasoning**, characterized by forward-chaining inference that follows a deductive logical progression from premises to conclusion, forming the essential backbone of solution development; **(2) Verification**, which represents metacognitive monitoring processes where previous calculations are systematically validated for accuracy, often using phrases like "Let me check"; **(3) Multi-method Validation**, demonstrating convergent thinking by applying diverse methodological approaches to reinforce conclusions, potentially introducing redundancy; and **(4) Error Correction** embodies a self-regulatory mechanism through which logical inconsistencies, computational errors, or potential mistakes are identified and remediated. This pattern captures the process of recognizing when a path of reasoning may be flawed, reassessing assumptions, and correcting course. While progressive reasoning constitutes the critical path to solution derivation, the other three functional patterns, though valuable in human-like reasoning, often contain redundancies that can be optimized without

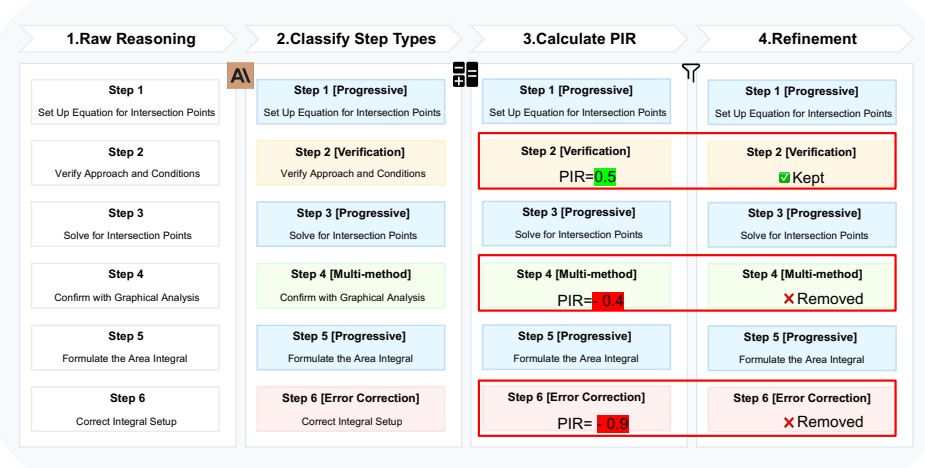

Figure 2: PIR framework pipeline for reasoning optimization: raw reasoning is segmented into logical steps, step is classified into reasoning patterns, PIR value is calculated to quantify step importance, and low-value functional steps are filtered while preserving progressive reasoning, resulting in more efficient reasoning chains.

compromising solution integrity. Details about the four reasoning patterns and corresponding cases can be found in Appendix A.

**PIR: Perplexity-Based Importance Refinement of Reasoning Steps**    PIR quantifies reasoning step importance by measuring perplexity changes when specific steps are removed. The indicator compares answer perplexity with and without a particular reasoning step:

$$\text{PIR}_\theta(x_i|x_{1:n}) = \log\left(\frac{\text{PPL}_\theta(R \setminus \{x_i\})}{\text{PPL}_\theta(R)}\right) \tag{1}$$

Where $\text{PPL}_\theta(R)$ and $\text{PPL}_\theta(R \setminus \{x_i\})$ represent perplexities calculated by model $\theta$ with and without step $i$:

$$\text{PPL}_\theta(R) = \exp\left(-\frac{1}{m}\sum_{j=1}^{m}\log p_\theta(a_j|x_{1:n}, a_{<j})\right) \tag{2}$$

$$\text{PPL}_\theta(R \setminus \{x_i\}) = \exp\left(-\frac{1}{m}\sum_{j=1}^{m}\log p_\theta(a_j|x_{1:i-1}, x_{i+1:n}, a_{<j})\right) \tag{3}$$

Here, $m$ represents the number of answer tokens, $a_j$ is the $j$-th answer token, $x_{1:n}$ represents all reasoning steps, and $x_{1:i-1}, x_{i+1:n}$ represents all steps except step $i$. The perplexity calculations are performed using a model $\theta$. A higher PIR value indicates greater step importance: when a critical reasoning step is eliminated, the model becomes significantly less confident in generating the correct answer, lacking essential information for solution derivation, resulting in higher perplexity and thus a higher PIR score.

## 2.3  Analysis and Optimization of Reasoning Chains

Our approach first employs a hierarchical decomposition process where reasoning chains are segmented into logical steps by Claude 3.7 Sonnet [1], with each step typically comprising multiple coherent sentences that form a cohesive reasoning unit. For classification, we implement a two-phase system: initially, a rule-based pattern matching component identifies steps containing characteristic linguistic markers (such as "Let me check" for verification or "I made a mistake" for error correction). For steps lacking explicit markers, we again apply Claude 3.7 Sonnet to perform contextual analysis, capturing more nuanced reasoning structures and assigning appropriate pattern classifications.

Table 1: Dataset Statistics. For each dataset, we report the data source, number of samples, total token count, and the distribution of tokens across four distinct reasoning patterns.

| Dataset | Source | Samples | Tokens | Progressive Reasoning | Verification | Multi-method Validation | Error Correction |
|---------|--------|---------|--------|----------------------|--------------|------------------------|------------------|
| S1K | Gemini | 1,000 | 4,509,505 | 71.4% | 9.2% | 13.9% | 5.5% |
| LIMO | DeepSeek-R1 | 817 | 5,144,004 | 59.7% | 11.8% | 10.9% | 17.6% |
| LIMO-V2 | QwQ | 800 | 8,866,950 | 64.3% | 9.8% | 12.7% | 13.2% |

To validate our classification methodology, we randomly select 5% of classified steps across our datasets for human evaluation. Four postgraduates independently assess whether each step is correctly classified according to our defined patterns. Steps are considered correctly classified only when all four annotators unanimously agree with the system's classification. This rigorous evaluation protocol revealed that 93.4% of the steps are unanimously verified as correctly classified, demonstrating the robust performance of our hybrid classification approach for reasoning step categorization.

Building on this classification system and the PIR metric, we implemented a targeted pruning approach that selectively identifies and removes low-importance functional steps while maintaining data quality and preserving reasoning integrity. This process is shown in Figure 2. Unlike approaches that might indiscriminately compress reasoning chains, our method specifically targets functional steps (verification, multi-method validation, and error correction) while preserving all progressive reasoning steps, which constitute the essential deductive core of the solution process. For each identified functional step, we compute its PIR value to quantify its importance based on the impact its removal has on answer prediction confidence. The process then selectively removes functional steps with the lowest PIR values according to a predefined ratio threshold, resulting in more concise reasoning chains that maintain effectiveness while significantly reducing verbosity.

This selective pruning approach creates more efficient reasoning chains that maintain the integrity of core problem-solving logic while eliminating redundant verification processes. The resulting optimized chains are both more efficient to process and more effective as training exemplars for downstream models, without sacrificing the quality and completeness of the essential reasoning. The Algorithm 1 in the appendix demonstrates this whole process. In our experiments, we apply Qwen2.5-32B-Instruct to calculate PPL.

## 3 Experimental Results and Analysis

### 3.1 Experimental Setup

**Training Dataset** To empirically validate our proposed PIR reasoning refinement framework, we conduct experiments using established datasets from prior work. As demonstrated in Table 1, our approach leverages three representative reasoning datasets: LIMO (distilled from DeepSeek R1), LIMO-V2 (distilled from QwQ), and S1K (distilled from Gemini Thinking). These datasets, distilled from different foundation models, represent diverse reasoning patterns and problem-solving approaches. Applying our framework across these varied sources allows us to validate the generalizability of our method across different data sources, ensuring that the PIR optimization is robust and not dependent on specific characteristics of any single source model. We apply our PIR framework to these datasets to create optimized versions: LIMO-P, LIMO-V2-P, and S1K-P. Additionally, we construct several variant datasets by implementing different refinement ratios to investigate the optimal balance between conciseness and effectiveness. Detailed information regarding different refinement ratios and the specific token counts of the optimized datasets is provided in the Appendix B.1.

**Benchmark Datasets** To rigorously assess our methodology, we utilized three representative reasoning-intensive benchmarks: **(1) AIME24** evaluation set encompasses 30 challenging problems from the American Invitational Mathematics Examination administered in early 2024, requiring sophisticated reasoning across mathematical domains; **(2) GPQA Diamond** corpus incorporates 198 doctoral-level scientific inquiries across biological, chemical, and physical disciplines, presenting formidable challenges that even subject matter experts struggle to master fully; **(3) AMC23** includes 40 problems from the AIMO progress prize competition. Using data after 2023.

**Performance metrics: Reasoning Effectiveness and Efficiency** To evaluate reasoning **effectiveness (ACC)**, we employ the pass@1 accuracy as our primary performance indicator. For each

Table 2: Experimental results comparing baseline models with their PIR-optimized variants (-P) across reasoning benchmarks. Metrics include accuracy (ACC), token length (TOK), and efficiency (EFF).

| Model | AIME | | | AMC | | | GPQA Diamond | | |
|---|---|---|---|---|---|---|---|---|---|
| | ACC ↑ | TOK ↓ | EFF ↑ | ACC ↑ | TOK ↓ | EFF ↑ | ACC ↑ | TOK ↓ | EFF ↑ |
| Qwen2.5-32B-Instruct | 15.8 | 954 | 1.66E-04 | 67.2 | 737 | 9.11E-04 | 47.0 | 517 | 9.08E-04 |
| R1-Distill-Qwen-32B | 69.2 | 9,311 | 7.43E-05 | 94.4 | 5,561 | 1.70E-04 | 64.7 | 5,634 | 1.15E-04 |
| QwQ | 81.7 | 12,234 | 6.68E-05 | 97.8 | 7,350 | 1.33E-04 | 70.2 | 7,483 | 9.38E-05 |
| S1-32B | 37.9 | 6,646 | 5.71E-05 | 80.9 | 4,542 | 1.78E-04 | 60.7 | 4,172 | 1.46E-04 |
| S1-32B-P | $42.1_{+4.2}$ | $4,716_{-29\%}$ | $8.92\text{E-}05_{+56\%}$ | $83.1_{+2.2}$ | $3,809_{-16\%}$ | $2.18\text{E-}04_{+22\%}$ | $61.6_{+0.9}$ | $2,472_{-41\%}$ | $2.49\text{E-}04_{+71\%}$ |
| LIMO | 56.7 | 12,497 | 4.53E-05 | 91.9 | 5,516 | 1.67E-04 | 67.2 | 7,173 | 9.36E-05 |
| LIMO-P | $63.3_{+6.6}$ | $10,588_{-15\%}$ | $5.98\text{E-}05_{+32\%}$ | $93.8_{+1.9}$ | $5,235_{-5\%}$ | $1.79\text{E-}04_{+7\%}$ | $71.2_{+4}$ | $6,969_{-3\%}$ | $1.02\text{E-}04_{+9\%}$ |
| LIMO-V2 | 66.3 | 13,896 | 4.77E-05 | 94.4 | 6,843 | 1.38E-04 | 70.2 | 8,035 | 8.74E-05 |
| LIMO-V2-P | $71.2_{+4.9}$ | $12,163_{-12\%}$ | $5.86\text{E-}05_{+23\%}$ | $96.6_{+2.2}$ | $6,348_{-7\%}$ | $1.52\text{E-}04_{+10\%}$ | $74.2_{+3}$ | $6,968_{-13\%}$ | $1.07\text{E-}04_{+22\%}$ |

problem in our benchmark, we sample eight responses from the model and calculate ACC under the Zero-shot Chain-of-Thought (CoT) setting with the instruction of: "Please reason step by step, and put your final answer within boxed." We utilize Qwen2.5-Math evaluators [35] to systematically assess solution correctness across all solutions, with each sampling conducted at a temperature setting of 0.7 to balance deterministic reasoning with exploration of solution paths. Building on our effectiveness measure, we quantify reasoning **efficiency (EFF)** as the ratio between model performance and resource utilization: EFF = ACC/TOK, where TOK represents the average number of response tokens across all benchmark problems. This efficiency metric captures the utility produced per unit of test-time resource consumption, effectively highlighting the critical trade-off between performance and computational cost.

**Evaluation of PIR Framework Across Multiple Training Datasets** We establish baseline models (LIMO, LIMO-V2, and S1-32B) that were trained on the original unmodified datasets using Qwen2.5-32B-Instruct [35]. These baseline models were obtained from their official Hugging Face repositories.[1] To test our PIR framework, we fine-tune the same Qwen2.5-32B-Instruct base model on the pruned datasets: LIMO-P, LIMO-V2-P, and S1K-P, using identical training scripts as described in the original papers and yielding our PIR-optimized models: LIMO-P, LIMO-V2-P, and S1-32B-P. All evaluations follow consistent testing protocols across methods to maintain comparative validity.

**Comparison Against Alternative Reasoning Optimization Methods** We compare our PIR framework with other competitive reasoning optimization methods. First, we establish **S1-32B** as our primary baseline trained on the original unmodified S1K dataset without any reasoning optimization. We then compare against two leading competing methods from prior work: Prompt Reduction [6], denoted as **S1-PROMPT**, which develops innovative prompting strategies that encourage LLMs to use shortcuts to quickly exploit reasoning clues and bypass detailed procedural steps. We apply this method for model S1-32B as a baseline; and SPIRIT [5], denoted as **S1-SPIRIT**, which applies a non-discriminative pruning approach for all reasoning steps based solely on perplexity values. We apply this method to the S1K dataset to get a pruned dataset, and fine-tuning Qwen2.5-32B-Instruct with the pruned dataset. Unlike our PIR method that selectively preserves essential progressive reasoning steps, S1-SPIRIT's uniform filtering risks removing critical components necessary for accurate solutions. Additionally, we include an ablation of our method, Rule-based Filtering (denoted as **S1-RULE**), which identifies functional step categories but randomly removes steps from these categories without using PIR metrics to determine their importance.[2]

---

[1]https://huggingface.co/

[2]We specifically chose S1K dataset for these comparisons because competing methods like SPIRIT require perplexity calculations for all sentences, which would be computationally prohibitive for the significantly larger LIMO or LIMO-V2 datasets, resulting in excessive computational costs and processing time.

Table 3: Experimental results comparing PIR(S1-32B-P) with different optimization approaches.

| Model/ Method | Training Tokens | AIME | | | AMC | | | GPQA Diamond | | |
|---|---|---|---|---|---|---|---|---|---|---|
| | | ACC ↑ | TOK ↓ | EFF ↑ | ACC ↑ | TOK ↓ | EFF ↑ | ACC ↑ | TOK ↓ | EFF ↑ |
| S1-32B | 4.51E+06 | 37.9 | 6,646 | 5.71E-05 | 80.9 | 4,542 | 1.78E-04 | 60.7 | 4,172 | 1.46E-04 |
| S1-PROMPT | 4.51E+06 | 36.7 | 8,013 | 4.58E-05 | 72.5 | 3,724 | 1.95E-04 | 58.0 | 2,853 | 2.03E-04 |
| S1-SPIRIT | 4.32E+06 | 37.1 | 4,906 | 7.56E-05 | 81.3 | 3,517 | 2.31E-04 | 60.1 | 2,818 | 2.13E-04 |
| S1-RULE | 4.31E+06 | 36.7 | 4,807 | 7.63E-05 | 81.3 | 3,654 | 2.22E-04 | 58.1 | 3,837 | 1.51E-04 |
| S1-32B-P | 4.31E+06 | $42.1_{+4.2}$ | $4,716_{-29\%}$ | $8.92E\text{-}05_{+56\%}$ | $83.1_{+2.2}$ | $3,809_{-16\%}$ | $2.18E\text{-}04_{+22\%}$ | $61.6_{+0.9}$ | $2,472_{-41\%}$ | $2.49E\text{-}04_{+71\%}$ |

## 3.2 Main Results

**Performance Across Benchmarks**  Table 2 demonstrates the consistent effectiveness of our PIR optimization framework across multiple challenging reasoning benchmarks. The PIR-optimized variants achieve superior efficiency-accuracy trade-offs across all model families. S1-32B-P shows remarkable improvements on AIME with a 4.2 percentage point accuracy increase alongside a 29% token reduction, yielding a 56% efficiency improvement. Similarly, LIMO-P demonstrates consistent gains with enhanced accuracy (+6.6, +1.9, and +4.0 percentage points across AIME, AMC, and GPQA Diamond, respectively) while reducing token consumption by up to 15%, achieving efficiency improvements of 32%, 7%, and 9% across the three benchmarks. LIMO-V2-P exhibits comparable enhancements with token reductions of 12%, 7%, and 13%, paired with accuracy improvements of 4.9, 2.2, and 3.0 percentage points. These consistent improvements across diverse benchmarks confirm that our PIR framework effectively identifies and preserves high-value reasoning steps while eliminating low-importance functional components.

**Generalizability to Different Data Sources**  The substantial performance improvements across models trained on data distilled from diverse foundation models (S1 from Gemini Flash Thinking, LIMO from DeepSeek-R1, and LIMO-V2 from QwQ) suggest PIR's strong generalizability potential. While individual gains vary in magnitude, the consistent pattern of simultaneous accuracy increases and token reductions across these different data sources indicates that our framework successfully identifies important reasoning patterns independent of the original distillation source. This suggests that PIR captures fundamental aspects of reasoning quality rather than exploiting source-specific characteristics.

**Comparison with Alternative Optimization Methods**  Table 3 demonstrates our PIR framework's superior performance compared to other reasoning optimization approaches. PIR (S1-32B-P) consistently outperforms all alternatives across benchmarks, achieving significant improvements over the baseline S1-32B model with accuracy gains of 4.2, 2.2, and 0.9 percentage points alongside token reductions of 29%, 16%, and 41% on AIME, AMC, and GPQA Diamond respectively. Notably, our approach of selectively pruning only functional steps while preserving all progressive reasoning components proves superior to whole-sentence pruning methods (S1-SPIRIT), which apply filtering indiscriminately across all reasoning steps. The superior accuracy of S1-32B-P over S1-SPIRIT, particularly on AIME (5.0 percentage point advantage), empirically validates our hypothesis that progressive reasoning steps constitute the essential solution backbone and should remain intact.

## 3.3 Analysis

### 3.3.1 Generalizability to Test-Time Scaling

Test-time scaling represents a critical dimension of LLM reasoning deployment, where more computational resources are allocated during inference. To evaluate whether our PIR optimization framework maintains its effectiveness across varying inference-time token budgets, we conducted experiments measuring accuracy as a function of token budget constraints. Figure 1 presents the test-time scaling curves across three benchmarks of LIMO and LIMO-P. The results demonstrate that PIR-optimized models consistently outperform their non-optimized counterparts across most token budget levels. The superior performance of PIR-optimized models across different token budgets demonstrates the generalizability of our approach to different resource constraints. This makes our PIR framework particularly valuable for real-world applications where inference efficiency and response latency are crucial considerations. The ability to maintain performance advantages highlights the robustness of our optimization methodology to different deployment scenarios.

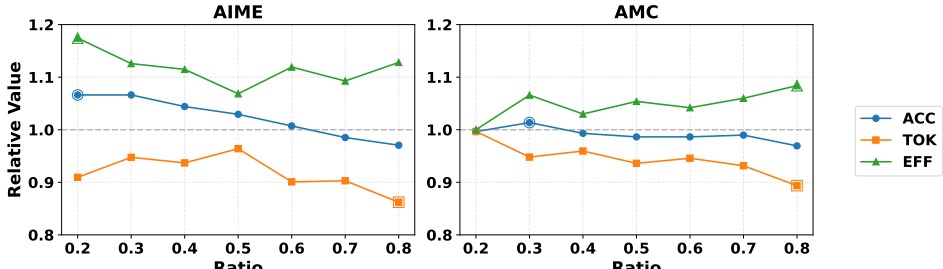

Figure 3: Impact of pruning ratio on model performance. This figure displays relative performance metrics (normalized to baseline) across different pruning ratios for AIME and AMC benchmarks. The horizontal dashed line represents the baseline performance (ratio=0).

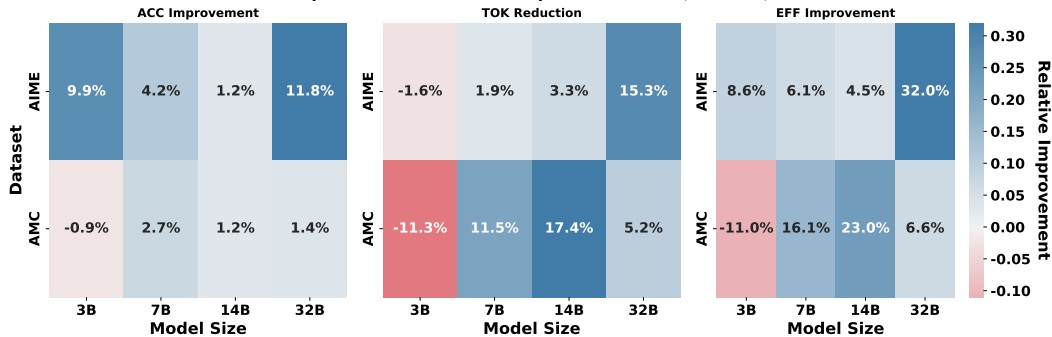

Figure 4: Impact of PIR refinement across model sizes and benchmarks. Heatmaps show relative percentage changes between models trained with pruned versus original datasets. Blue indicates improvement: higher accuracy, shorter response length, or better efficiency.

### 3.3.2 Impact of ratios

We investigated the impact of various pruning ratios on model performance by creating multiple LIMO dataset variants with different proportions of functional reasoning steps removed. As shown in Figure 3, our experiments revealed clear performance trade-offs across metrics. For the AIME benchmark, lower pruning ratios (0.2-0.3) yielded optimal accuracy improvements over the baseline, while higher ratios (0.8) achieved the greatest response length reduction. For AMC, a moderate pruning ratio of 0.3 delivered peak accuracy while maintaining efficiency gains. Test time efficiency consistently improved with pruning across both benchmarks, with particularly strong gains on AIME. These results demonstrate the existence of an optimal refinement threshold that balances the removal of redundant functional steps with the preservation of critical reasoning components. Excessive pruning beyond this threshold leads to declining accuracy as valuable reasoning elements are removed, even when categorized as functional steps. This pattern validates our PIR framework's approach of selectively identifying and preserving high-value reasoning steps based on their quantitative importance scores.

### 3.3.3 Generalizability to Model Size

To evaluate PIR's generalizability across parameter scales, we conducted experiments using Qwen2.5 models ranging from 3B to 32B parameters, comparing the performance of models trained with optimized LIMO-P versus original LIMO datasets. As shown in Figure 4, our method demonstrates robust scalability with performance improvements across most model sizes. The benefits of PIR refinement become increasingly pronounced as model size increases, particularly for the AIME benchmark where the 32B model shows impressive gains across all metrics (11.8% accuracy improvement, 15.3% response length reduction, and 32.0% efficiency increase). For AMC, mid-sized models (7B-14B) yield the strongest efficiency improvements (up to 23.0%). The consistent pattern of enhancement for most model sizes suggests that our method's scalability and practical utility across various model sizes.

# 4 Related Work

## 4.1 Test-Time Scaling of LLMs

Test-time scaling [12, 26, 29, 10] enhances LLM reasoning by increasing inference-time computation. Approaches include non-training methods, which optimize existing model inference strategies, and training-based methods, which modify model parameters. Non-training techniques encompass Best-of-N sampling [18, 27], majority voting [31, 3], and tree search [36, 30] for exploring multiple reasoning paths. Training-based approaches divide into supervised fine-tuning (SFT) and reinforcement learning (RL). SFT methods train on high-quality reasoning traces [20, 37, 16, 15], with S1 and LIMO demonstrating improved performance through careful sample selection. RL approaches [24, 10, 12, 29] have yielded exceptional results, with DeepSeek-R1 using GRPO and OpenAI's o1 and QwQ enabling autonomous development of reasoning chains that adaptively allocate computation based on problem complexity. Our work focuses on optimizing SFT-based approaches through perplexity-based refinement to improve efficiency while preserving accuracy.

## 4.2 Efficient Reasoning

Research on efficient reasoning has gained significant traction as LLMs face challenges with computational overhead and verbosity. At inference time, various optimization approaches have emerged without requiring parameter updates. Length budgeting techniques like Token-Budget-Aware LLM Reasoning [11] enforce token limits via prompting, while S1 [20] appends end-of-thinking delimiters. Dual-process inspired system switching methods alternate between fast intuitive and deliberative reasoning; Dualformer [28] selectively drops reasoning traces during training, while System 1.x [23] employs a controller to assess task difficulty. Model switching approaches such as BiLD [14] and EAGLE [17] leverage speculative decoding with smaller models for initial predictions, while RouteLLM [21] routes queries based on complexity. For supervised fine-tuning, C3ot [13] preserves essential information while reducing redundancies, and TokenSkip [34] omits less important tokens. SPIRIT [5] prunes low-importance reasoning steps. RL-based methods either incorporate explicit length penalties [19, 25] or balance exploitation with exploration [22]. Most closely related to our work, SPIRIT [5] calculates perplexity for all reasoning steps and filters them based on a predetermined ratio. Our approach fundamentally differs by distinguishing between reasoning step types rather than treating all equally. We preserve all progressive reasoning steps—the essential solution backbone—while only pruning less critical functional components (verification, multi-method validation, and error correction), ensuring core reasoning integrity and avoiding the risk of removing critical solution elements.

# 5 Conclusion

**Contributions** This paper introduces PIR (Perplexity-based Importance Refinement), a novel framework that optimizes reasoning chains by quantitatively assessing step importance and selectively pruning low-value functional elements while preserving essential reasoning paths. Our comprehensive evaluation demonstrates that models fine-tuned on PIR-optimized datasets achieve both improved accuracy and significantly reduced token usage. By strategically balancing thorough problem-solving with computational efficiency, PIR establishes a principled approach for deploying advanced reasoning capabilities in latency-sensitive applications, opening new avenues for research on efficient reasoning in foundation models.

**Limitations** While our approach demonstrates significant improvements, several limitations warrant further investigation. First, our evaluation primarily focuses on mathematical reasoning tasks and science tasks; future work should validate PIR's effectiveness across broader reasoning domains including logical, commonsense, and causal reasoning. Second, our refinement strategy relies on perplexity as the primary importance indicator, which may not fully capture the semantic contribution of certain reasoning steps. Alternative metrics incorporating semantic relevance could enhance refinement precision. Third, the optimal pruning ratio may vary across different reasoning tasks and model architectures, suggesting the need for adaptive refinement strategies. Finally, our approach currently requires access to model perplexity outputs, which may limit applicability with closed-source models. Addressing these limitations could further advance efficient reasoning frameworks for real-world applications.

# 6 Acknowledgements

We would like to thank all reviewers for their insightful comments and suggestions to help improve the paper. This work was partially supported by the Research Grants Council of Hong Kong (GRF No. 15209724). This work was also partially funded by the National Natural Science Foundation of China (62476168) and SII.

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

# A   Reasoning Refinement: Reasoning Optimization Framework

## A.1   Phrases Used for Pattern Matching

In our rule-based pattern matching system for reasoning step classification, we identify four distinct cognitive reasoning patterns characterized by specific linguistic markers. Table 4 presents these patterns alongside their characteristic phrases. Figure 5 demonstrates one sample from S1K containing the four patterns.

**Progressive Reasoning** constitutes the critical path to solution derivation, characterized by forward-chaining inference that follows a deductive logical progression from premises to conclusion. This pattern forms the essential backbone of solution development and is indicated by phrases that signal logical advancement, such as "Let's solve," "First/Then/Next," and "Therefore."

**Verification** represents metacognitive monitoring processes where previous calculations are systematically validated for accuracy. This pattern is identified through phrases like "Let me check," "Let me verify," and "Double-check," which signal when the model reviews its prior work.

**Multi-method Validation** demonstrates convergent thinking through the application of diverse methodological approaches to reinforce conclusions. This pattern is recognized through expressions indicating alternative approach consideration, including "Alternatively," "Another way," and "Let's try a different approach," potentially introducing redundancy.

**Error Correction** embodies a self-regulatory mechanism through which logical inconsistencies or computational errors are identified and remediated. This pattern is captured by phrases acknowledging mistakes, such as "This is wrong," "The mistake was," and "This contradicts."

Table 4: Cognitive Reasoning Patterns and Associated Phrases

| Cognitive Reasoning Patterns | Phrases |
| --- | --- |
| Progressive Reasoning | "Let's solve"
"First/Then/Next"
"Therefore"
"We need to"
"Given that" |
| Verification | "Wait"
"Let me check"
"Let me verify"
"Double-check"
"Going back to" |
| Multi-method Validation | "Alternatively"
"Another way"
"Let's try a different approach"
"Using another method"
"We can also verify" |
| Error Correction | "This is wrong"
"The mistake was"
"That's impossible"
"This contradicts"
"The error is" |

## A.2   Prompt used for Pattern Recognition

Our hybrid classification system combines rule-based pattern matching with contextual analysis using Claude 3.7 Sonnet for more nuanced reasoning pattern recognition. In Figure 6 and 7, we present the prompts used for the sophisticated pattern recognition of Claude 3.7 Sonnet.

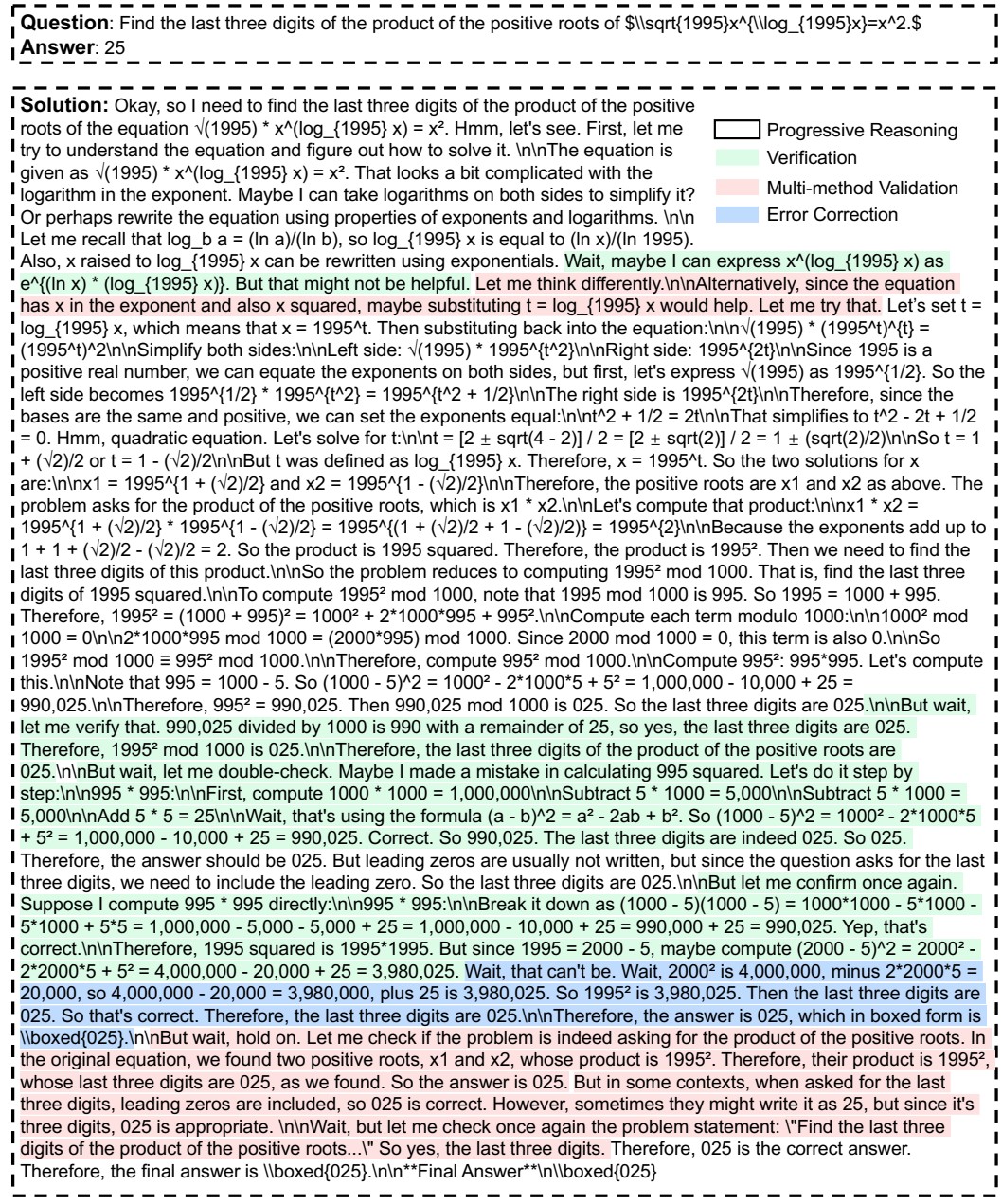

Figure 5: A case where one training sample contains the four patterns.

## A.3 The Pseudocode of the Pipeline of PIR

Algorithm 1 outlines the complete pipeline of our Perplexity-based Importance Refinement (PIR) framework. Our PIR framework implements a streamlined three-phase process for optimizing reasoning chains. First, Claude 3.7 Sonnet performs hierarchical decomposition of reasoning into coherent steps, which are classified using our hybrid system combining rule-based pattern matching and contextual analysis. Second, we quantify each functional step's importance by calculating its PIR value—the logarithmic ratio between the perplexity of generating the answer with and without that step using Qwen2.5-32B-Instruct. Finally, we apply pattern-specific selective pruning, preserving all progressive reasoning while removing only low-value functional components (verification, multi-method validation, and error correction) based on their PIR scores, thereby maintaining solution integrity while significantly reducing verbosity.

"# Mathematical Reasoning Analysis JSON Prompt\n\nAnalyze the provided mathematical reasoning solution by identifying the thinking patterns into a structured JSON format.\n\n## Pattern Definitions and Examples\n\n### 1. Progressive Reasoning\n\n**Definition**: The standard forward-moving problem-solving process that follows logical order.\n\n**Key Indicators**: \"Let's solve\", \"First/Then/Next\", \"Therefore\", \"We need to\", \"Given that\"\n\n**Example**:\n\n```\n\"Let's solve this step by step:\"\n\"First, we need to calculate n = 2^31 * 3^19\"\n\"So, we start by calculating this value.\"\n\"Now, let's compute the value of n.\"\n\"Therefore, let's find n = 2^31 * 3^19.\"\n\n```\n\n### 2. Verification\n\n**Definition**: Process of returning to check previous steps for accuracy.\n\n**Key Indicators**: \"Wait\", \"Let me check\", \"Let me verify\", \"Double-check\", \"Going back to\"\n\n**Example**:\n\n```\n\"Wait, let me check the multiplication:\"\n\"32 * 20 = 640 √\"\n\"Let me verify this calculation:\"\n\"32 * 20 = 640 √\"\n\"Just to be extra sure:\"\n\"32 * 20 = 640 √\"\n\"We got 640, which looks right.\"\n\n```\n\n### 3. Multi-method Validation\n\n**Definition**: Using different methods or perspectives to verify a conclusion.\n\n**Key Indicators**: \"Alternatively\", \"Another way\", \"Let's try a different approach\", \"Using another method\", \"We can also verify\"\n\n**Example**:\n\n```\n\"Let's verify with n = 6:\"\n\"[calculation steps with n = 6]\"\n\"Alternatively, let's try n = 12:\"\n\"[calculation steps with n = 12]\"\n\"Let's also check n = 24:\"\n\"[calculation steps with n = 24]\"\n\"And verify with n = 36:\"\n\"[calculation steps with n = 36]\"\n\n```\n\n### 4. Error Correction Pattern\n\n**Definition**: Process of identifying and correcting mistakes in reasoning.\n\n**Key Indicators**: \"This is wrong\", \"The mistake was\", \"That's impossible\", \"This contradicts\", \"The error is\"\n\n**Example**:\n\n```\n\"This can't be right because the total is odd.\"\n\"The mistake is in assuming all pairs are non-divisors.\"\n\"Wait, I made a mistake in the pairing assumption.\"\n\"Let me correct this pairing error.\"\n\"The problem was in how we thought about pairs.\"\n\"We need to fix our understanding of pairs.\"\n\n```\n\n## Analysis Requirements\n\n1. **Group Related Sentences**: Identify related sentences that together form coherent sub-thinking processes.\n2. **Assign IDs**: Each sub-thinking process should have its own unique ID.\n3. **Structure Output as JSON**: Return your analysis in JSON format with:\n   - A dictionary with all sentences separated into sub-groups\n4. **Group Size Limitation**: Each sentence group should contain a reasonable number of sentences (typically 3-4) that represent one coherent step or thought. Avoid creating overly large groups that combine multiple distinct steps or thoughts.\n\n## Expected JSON Format\n\n```json\n{\n   \"sentence_groups\": {\n     \"group1\": [\"sentence1\", \"sentence2\", \"...\"],\n     \"group2\": [\"sentence1\", \"sentence2\", \"...\"],\n     \"...\": \"...\"\n   }\n}\n\n```\n\nFocus on identifying complete thinking processes rather than labeling individual sentences. Each group should represent a coherent step or thought in the mathematical reasoning. Do not delete or modify any sentences!!! Please just group the specified reasoning process."

Figure 6: The prompt to segment coherent sub-thinking sentences into cohesive reasoning steps.

# B Experiments

## B.1 Dataset Statistics

This section provides comprehensive statistics for the datasets used in our experiments before and after applying the PIR optimization framework at different thresholds. Table 5 presents the number of examples and total token counts for each dataset variant. The original datasets (S1, LIMO, and LIMO-V2) were derived from three different Large Reasoning Models: Gemini Flash Thinking, DeepSeek-R1, and QwQ, respectively. Each dataset was then processed using our PIR framework at varying optimization ratios (from 0.2 to 0.8), where higher values indicate more aggressive pruning of functional reasoning steps.

## B.2 Main Results with Training Tokens

Table 6 presents a comprehensive evaluation of our PIR optimization approach across three challenging benchmarks: AIME (American Invitational Mathematics Examination), AMC (American Mathematics Competition), and GPQA Diamond. We include training token counts alongside accuracy, response length, and test-time efficiency to highlight the relationship between data efficiency and model performance. The baseline models—R1-Distill-Qwen-32B, Qwen2.5-32B-Instruct, and QWQ—establish performance references across the benchmarks. QWQ demonstrates superior accuracy but requires substantially longer responses, which impacts its test-time efficiency. Qwen2.5-32B-Instruct offers the highest efficiency due to its concise responses, albeit with lower accuracy. For our optimized models, we observe consistent patterns across all three dataset families (S1, LIMO, and LIMO-V2): **S1 datasets:** The PIR-optimized variant (S1-32B-P) achieves higher accuracy on AIME (+4.2%) and AMC (+2.2%) while using 4.4% fewer training tokens. The average response length decreases by 29% for AIME and 16% for AMC, resulting in efficiency improvements of 56% and 22%, respectively. **LIMO datasets:** The PIR-optimized model (LIMO-P) demonstrates accuracy improvements across all benchmarks (+6.6% on AIME, +1.9% on AMC, +4% on GPQA) while requiring 8.8% fewer training tokens. Response length reductions of 15% for AIME and 5% for AMC translate to efficiency gains of 32% and 7%, respectively. **LIMO-V2 datasets:** PIR optimization (LIMO-V2-P) achieves the consistent accuracy improvements (+4.9% on AIME, +2.2% on AMC, +3% on GPQA) while using 5.3% fewer training tokens. Response lengths decrease

"# Mathematical Reasoning Analysis JSON Prompt\n\nAnalyze the provided mathematical reasoning solution by categorizing the subprocess of the reasoning solution to thinking patterns into a structured JSON format.\n\n## Pattern Definitions and Examples\n\n### 1. Progressive Reasoning\n\n**Definition**: The standard forward-moving problem-solving process that follows logical order.\n\n**Key Indicators**: \"Let's solve\", \"First/Then/Next\", \"Therefore\", \"We need to\", \"Given that\"\n\n**Example**:\n\n```\n\"Let's solve this step by step:\"\n\"First, we need to calculate n = 2^31 * 3^19\"\n\"So, we start by calculating this value.\"\n\"Now, let's compute the value of n.\"\n\"Therefore, let's find n = 2^31 * 3^19.\"\n\n```\n\n### 2. Verification\n\n**Definition**: Process of returning to check previous steps for accuracy.\n\n**Key Indicators**: \"Wait\", \"Let me check\", \"Let me verify\", \"Double-check\", \"Going back to\"\n\n**Example**:\n\n```\n\"Wait, let me check the multiplication:\"\n\"32 * 20 = 640 ✓\"\n\"Let me verify this calculation:\"\n\"32 * 20 = 640 ✓\"\n\"Just to be extra sure:\"\n\"32 * 20 = 640 ✓\"\n\"We got 640, which looks right.\"\n\n```\n\n### 3. Multi-method Validation\n\n**Definition**: Using different methods or perspectives to verify a conclusion.\n\n**Key Indicators**: \"Alternatively\", \"Another way\", \"Let's try a different approach\", \"Using another method\", \"We can also verify\"\n\n**Example**:\n\n```\n\"Let's verify with n = 6:\"\n\"[calculation steps with n = 6]\"\n\"Alternatively, let's try n = 12:\"\n\"[calculation steps with n = 12]\"\n\"Let's also check n = 24:\"\n\"[calculation steps with n = 24]\"\n\"And verify with n = 36:\"\n\"[calculation steps with n = 36]\"\n\n```\n\n### 4. Error Correction Pattern\n\n**Definition**: Process of identifying and correcting mistakes in reasoning.\n\n**Key Indicators**: \"This is wrong\", \"The mistake was\", \"That's impossible\", \"This contradicts\", \"The error is\"\n\n**Example**:\n\n```\n\"This can't be right because the total is odd.\"\n\"The mistake is in assuming all pairs are non-divisors.\"\n\"Wait, I made a mistake in the pairing assumption.\"\n\"Let me correct this pairing error.\"\n\"The problem was in how we thought about pairs.\"\n\"We need to fix our understanding of pairs.\"\n\n```\n\n## Analysis Requirements\n\n1. **Structure Output as JSON**: Return your analysis in JSON format with:\n- A dictionary mapping pattern types to lists of sub-thinking process IDs\n\n2. Each sentence group should be categorized as either Regular Reasoning OR one of the other patterns, but never both. Each sentence group should be categorized as one of the four patterns.\n\n3. **Pattern Identification**:\n    - **Progressive Reasoning**: One continuous pattern from start to finish with logical steps\n    - **Verification**: Moments where solver checks previous work (reference which Regular Reasoning ID is being verified)\n    - **Multi-method Validation**: Separate different verification methods, each with its own ID (reference which Regular Reasoning ID is being checked with multi methods)\n    - **Error Correction Pattern**: Instances where mistakes are caught and fixed (reference which Regular Reasoning ID is being corrected)\n\n## Expected JSON Format\n\n```json\n{\n \"patterns\": {\n \"regular_reasoning\": [\"group1\", \"group3\", \"...\"],\n \"backtracking_verification\": [ {\n \"id\": \"group2\",\n \"relates_to\": \"group1\"\n } ],\n \"multiple_verification\": [ {\n \"id\": \"group4\",\n \"relates_to\": \"group1\",\n \"method\": \"Method Description\"\n },\n {\n \"id\": \"group5\",\n \"relates_to\": \"group1\",\n \"method\": \"Method Description\"\n } ],\n \"error_correction\": [\n {\n \"id\": \"group6\",\n \"relates_to\": \"group1\"\n } ]\n }\n}\n\n```\n"

Figure 7: The prompt to categorize the steps into reasoning patterns

consistently (12% for AIME, 7% for AMC, 13% for GPQA), yielding efficiency improvements of 23%, 10%, and 22%, respectively. These results demonstrate that our PIR framework effectively reduces training token requirements while simultaneously improving both accuracy and efficiency across diverse reasoning tasks. The consistent performance improvements across different model families validate the generalizability of our approach. Notably, the LIMO-V2-P model achieves state-of-the-art performance on all benchmarks while maintaining competitive efficiency, highlighting the effectiveness of optimizing reasoning chains by preserving essential progressive reasoning while removing less valuable functional steps.

## B.3 Impact of Cognitive Reasoning Patterns

Our empirical analysis of cognitive reasoning patterns reveals distinct performance characteristics across the evaluated benchmarks. As shown in Figure 8, the four identified reasoning patterns exhibit different trade-offs between accuracy and computational efficiency. Progressive Reasoning (PR) provides a solid baseline, while PR+Error Correction demonstrates the most balanced performance improvement on several datasets, delivering notable accuracy gains with competitive or even improved efficiency. These findings validate our PIR framework's approach of quantitatively evaluating reasoning step importance to identify and selectively preserve high-value steps while pruning those with minimal contribution, creating optimized reasoning chains that balance accuracy gains with computational demands. The observed performance variations highlight that different reasoning patterns contribute differentially to model performance, with certain patterns delivering more substantial benefits in specific contexts. This suggests that selectively preserving the most valuable functional reasoning components while removing redundant steps can effectively optimize the efficiency-accuracy trade-off in reasoning chains.

**Algorithm 1:** Reasoning Chain Optimization via PIR

**Input:** $solution$: original reasoning chain, $answer$: solution answer, $ratio$: pruning threshold, $\theta$: evaluation model

**Output:** $solution_{opt}$: optimized reasoning chain

```
// Step 1:  Segment and classify reasoning chain steps
```
$steps \leftarrow$ SegmentIntoLogicalSteps($solution, \theta$);
$classified\_steps \leftarrow$ ClassifyReasoningPatterns($steps, \theta$);
$functional\_steps \leftarrow$ FilterByPatterns($classified\_steps$, {Verification, Multi-method, Error correction});

```
// Step 2:  Calculate baseline perplexity with complete reasoning
```
$PPL_\theta(R) \leftarrow$ CalculatePerplexity($solution, answer, \theta$);

```
// Step 3:  Evaluate importance of each functional step
```
**foreach** $step_i \in functional\_steps$ **do**
    $solution_{-i} \leftarrow$ RemoveStep($solution, step_i$);
    $PPL_\theta(R \setminus \{step_i\}) \leftarrow$ CalculatePerplexity($solution_{-i}, answer, \theta$);
    $step_i.PIR \leftarrow \log\left(\frac{PPL_\theta(R \setminus \{step_i\})}{PPL_\theta(R)}\right)$;
**end**

```
// Step 4:  Selectively prune low-importance steps by pattern type
```
$solution_{opt} \leftarrow solution$;
**foreach** $pattern \in \{Verification, Multi\text{-}method, Error\ correction\}$ **do**
    $pattern\_steps \leftarrow$ FilterByPattern($functional\_steps, pattern$);
    $threshold \leftarrow$ CalculatePruningThreshold($pattern\_steps, ratio$);
    $steps\_to\_prune \leftarrow$ SelectLowPIRSteps($pattern\_steps, threshold$);
    $solution_{opt} \leftarrow$ RemoveSteps($solution_{opt}, steps\_to\_prune$);
**end**
**return** $solution_{opt}$;

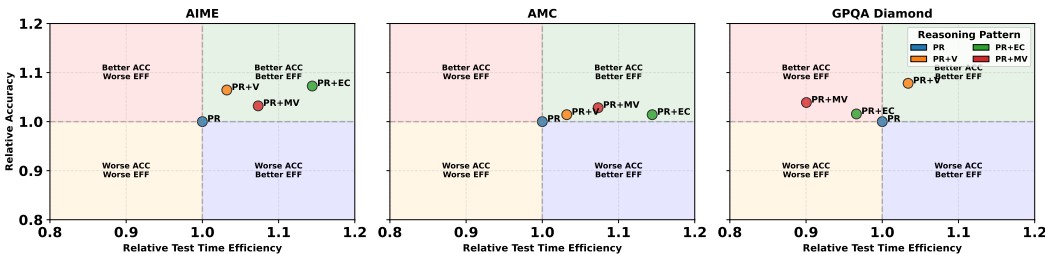

Figure 8: Impact of reasoning patterns on model performance across different benchmarks. Each subplot displays the relative accuracy (y-axis) versus relative test time efficiency (x-axis) compared to the Progressive Reasoning (PR) baseline. PR represents the model trained with only progressive reasoning steps. PR+Verification (PR+V) is trained with the dataset that includes progressive reasoning and verification steps. PR+Error Correction (PR+EC) stands for the model trained with progressive reasoning and error correction steps. PR+Multi-method Validation (PR+MV) is trained with progressive reasoning and multi-method validation steps.

## B.4 Case Study

To illustrate the effectiveness of our Perplexity-based Importance Refinement (PIR) framework, we present a detailed case study examining how models trained on refined reasoning chains differ in their inference behavior. Figure 9 provides a side-by-side comparison of responses generated by two models: one trained on the original LIMO dataset (left) and another trained on our PIR-optimized LIMO-P dataset (right), when presented with an identical mathematical problem. The model trained on original LIMO data exhibits characteristically verbose reasoning patterns inherited from its training data (left panel, 3,234 tokens). Despite reaching the correct answer, this model produces extensive verification steps, redundant calculations, and multiple self-checking procedures. The response includes numerous instances of recalculation, approach reassessment, and duplicate validations—reflecting the verbose nature of the LRM-distilled training data it was fine-tuned on. In striking contrast, the model trained on PIR-optimized data produces a significantly more concise response while maintaining solution accuracy (right panel, 1,612 tokens). This model has learned to focus on essential progressive reasoning pathways while minimizing unnecessary verification steps. The 50% reduction in output token count demonstrates that models trained on PIR-refined data effectively internalize more efficient reasoning strategies without compromising problem-solving

Table 5: Statistics of Dataset Variants after PIR Optimization

| Source | Data | Numbers | Tokens |
|---|---|---|---|
| Gemini | S1 | 1000 | 4509505 |
| | S1-0.2 | 1000 | 4440447 |
| | S1-0.3 | 1000 | 4390349 |
| | S1-0.4 | 1000 | 4307878 |
| | S1-0.5 | 1000 | 4173726 |
| | S1-0.6 | 1000 | 4136261 |
| | S1-0.7 | 1000 | 4064908 |
| | S1-0.8 | 1000 | 3998891 |
| DeepSeek-R1 | LIMO | 817 | 5144004 |
| | LIMO-0.2 | 817 | 4971633 |
| | LIMO-0.3 | 817 | 4865402 |
| | LIMO-0.4 | 817 | 4724104 |
| | LIMO-0.5 | 817 | 4542101 |
| | LIMO-0.6 | 817 | 4459545 |
| | LIMO-0.7 | 817 | 4342583 |
| | LIMO-0.8 | 817 | 4220217 |
| QwQ | LIMO-V2 | 800 | 8866950 |
| | LIMO-V2-0.2 | 800 | 8488260 |
| | LIMO-V2-0.3 | 800 | 8398975 |
| | LIMO-V2-0.4 | 800 | 8292603 |
| | LIMO-V2-0.5 | 800 | 8161582 |
| | LIMO-V2-0.6 | 800 | 8082063 |
| | LIMO-V2-0.7 | 800 | 7980495 |
| | LIMO-V2-0.8 | 800 | 7877110 |

Table 6: Performance comparison across different models on AIME, AMC, and GPQA Diamond benchmarks.

| Model | AIME | | | | AMC | | | | GPQA Diamond | | | |
|---|---|---|---|---|---|---|---|---|---|---|---|---|
| | Training Tokens | Acc | Avg. Response Tokens | Test Time Efficiency | Training Tokens | Acc | Avg. Response Tokens | Test Time Efficiency | Training Tokens | Acc | Avg. Response Tokens | Test Time Efficiency |
| Qwen2.5-32B-Instruct | N/A | 15.8 | 954 | 1.66E-04 | N/A | 67.2 | 737 | 9.11E-04 | N/A | 47.0 | 517 | 9.08E-04 |
| R1-Distill-Qwen-32B | N/A | 69.2 | 9,311 | 7.43E-05 | N/A | 94.4 | 5,561 | 1.70E-04 | N/A | 64.7 | 5,634 | 1.15E-04 |
| QWQ | N/A | 81.7 | 12,234 | 6.68E-05 | N/A | 97.8 | 7,350 | 1.33E-04 | N/A | 70.2 | 7,483 | 9.38E-05 |
| S1 | | | | | | | | | | | | |
| S1-32B | 4.51E+06 | 37.9 | 6,646 | 5.71E-05 | 4.51E+06 | 80.9 | 4,542 | 1.78E-04 | 4.51E+06 | 60.7 | 4,172 | 1.46E-04 |
| S1-32B-P | 4.31E+06 | 42.1 | 4,716 | 8.92E-05 | 4.31E+06 | 83.1 | 3,809 | 2.18E-04 | 4.39E+06 | 61.6 | 2,472 | 2.49E-04 |
| LIMO | | | | | | | | | | | | |
| LIMO | 5.14E+06 | 56.7 | 12,497 | 4.53E-05 | 5.14E+06 | 91.9 | 5,516 | 1.67E-04 | 5.14E+06 | 67.2 | 7,173 | 9.36E-05 |
| LIMO-P | 4.69E+06 | 63.3 | 10,588 | 5.98E-05 | 4.78E+06 | 93.8 | 5,235 | 1.79E-04 | 4.72E+06 | 71.2 | 6,969 | 1.02E-04 |
| LIMO-V2 | | | | | | | | | | | | |
| LIMO-V2 | 8.87E+06 | 66.3 | 13,896 | 4.77E-05 | 8.87E+06 | 94.4 | 6,843 | 1.38E-04 | 8.87E+06 | 70.2 | 8,035 | 8.74E-05 |
| LIMO-V2-P | 8.40E+06 | 71.2 | 12,163 | 5.65E-05 | 8.40E+06 | 96.6 | 6,348 | 1.52E-04 | 8.49E+06 | 74.2 | 6,968 | 1.07E-04 |

capabilities. Qualitative analysis of both responses reveals that while the token count differs substantially, both models arrive at the correct solution (204 minutes). However, the PIR-trained model achieves this with greater efficiency, focusing on core mathematical operations and direct solution paths rather than extensive self-verification. This confirms that models inherit the reasoning patterns present in their training data—when trained on strategically refined examples, they learn to generate more streamlined responses. This case study highlights a critical advantage of our approach: by optimizing training data through selective removal of low-importance functional steps (based on our PIR metric), we influence downstream model behavior to favor efficiency without sacrificing effectiveness. The PIR-trained model demonstrates that the extensive verification processes prevalent in LRM reasoning—while mimicking human problem-solving—can be substantially reduced in practical applications without compromising solution quality.

**Question**: "Every morning Aya goes for a $9$-kilometer-long walk and stops at a coffee shop afterwards. When she walks at a constant speed of $s$ kilometers per hour, the walk takes her 4 hours, including $t$ minutes spent in the coffee shop. When she walks $s+2$ kilometers per hour, the walk takes her 2 hours and 24 minutes, including $t$ minutes spent in the coffee shop. Suppose Aya walks at $s+\frac{1}{2}$ kilometers per hour. Find the number of minutes the walk takes her, including the $t$ minutes spent in the coffee shop."

**Answer**: 204

---

**LIMO** — Token:3234

Okay, let's see. I need to solve this problem where Aya is walking different distances at different speeds, and we have to find out how long her walk takes when she walks at s + 1/2 km/h, including the time t she spends in the coffee shop. <Omitted>\n\nWait, 11.25 is equal to 22.5 / 2? Wait, no. Wait, 11.25 is 45/4. Hmm, better to keep it as fractions perhaps.\n\nAlternatively, maybe work with decimals here. Let me see.\n\nSo, quadratic equation: s² + 2s - 11.25 = 0\n\nLet me multiply both sides by 4 to eliminate decimals:\n\n4s² + 8s - 45 = 0\n\nNow, quadratic equation: 4s² + 8s - 45 = 0\n\nUsing quadratic formula:\n\ns = [ -b ± sqrt(b² - 4ac) ]/(2a)\n\nHere, a=4, b=8, c=-45\n\nDiscriminant D = b² -4ac = (8)^2 - 4*4*(-45) = 64 + 720 = 784\n\nsqrt(784)=28\n\nTherefore,\n\ns = [ -8 ± 28 ] / (2*4) = (-8 ±28)/8\n\nTaking the positive root since speed can't be negative:\n\ns = (-8 +28)/8 = 20/8 = 2.5 km/h\n\nSo, s = 2.5 km/h\n\nNow, let's verify this value. Let me check if substituting s=2.5 into the equations gives consistent results.\n\nFirst equation: 9/s + t/60 =4\n\n9/2.5 = 3.6\n\nSo 3.6 + t/60 =4 → t/60 =0.4 → t=24 minutes\n\nSecond equation: 9/(s+2) + t/60 =2.4\n\n\ns+2=4.5 km/h\n\n9/4.5 =2 hours\n\nSo 2 + t/60 =2.4 → t/60=0.4 → t=24 minutes, which matches.\n\nSo, s=2.5 km/h and t=24 minutes.\n\nNow, the question asks: If Aya walks at s + 1/2 km/h, find the number of minutes the walk takes when she walks at the coffee shop.\n\nFirst, compute s + 1/2:\n\ns =2.5 km/h, so s + 0.5 = 3 km/h\n\nWalking speed is 3 km/h. The distance is still 9 km, so walking time is 9 /3 = 3 hours.\n\nIncluding the coffee time t=24 minutes, total time is 3 hours +24 minutes.\n\nConvert 3 hours to minutes: 3*60=180 minutes, plus 24 minutes gives 204 minutes.\n\nWait, but the question says \"the walk takes her\", does that include the coffee time or not? Wait, the problem says: \"Find the number of minutes the walk takes her, including the t minutes spent in the coffee shop.\"\n\nAh, so the total time is the walking time plus t minutes. So yes, exactly as above.\n\nSo total time is 3 hours walking +24 minutes coffee, which is 3*60 +24 = 180 +24=204 minutes.\n\nWait, but let me double-check my steps to make sure I didn't make any miscalculations.\n\nFirst, solving for s and t:\n\nFrom the first scenario:\n\nWalking at 2.5 km/h, walking time is 9/2.5 = 3.6 hours. Then total time is 3.6 + t/60 =4, so t/60=0.4 → t=24 minutes. Correct.\n\nSecond scenario: walking at 2.5+2=4.5 km/h, walking time 9/4.5=2 hours. Total time 2 + t/60 =2.4 → t=24 minutes. Correct.\n\nThen, walking at s + 1/2 =2.5 +0.5=3 km/h. Walking time is 9/3=3 hours. Total time is 3 hours +24 minutes, which is 3*60 +24=180+24=204 minutes.\n\nYes, that seems correct.\n\nBut just to be thorough, let me check if there was another way someone might misinterpret the problem.\n\nFor example, maybe the problem says \"the walk takes her 4 hours\" including the coffee shop time, so perhaps the walking time is 4 hours minus t minutes. Wait, actually, that's exactly how we modeled it. The walking time is distance over speed, then add the coffee time to get the total time. So our model is correct.\n\nAlternatively, maybe someone might think that the \"walk takes her\" refers only to the walking time, but the problem specifically says \"including t minutes spent in the coffee shop\". So the total time is walking time plus t. Therefore, the answer is indeed 204 minutes.\n\nWait, but let me confirm once more. The problem states that in the first scenario, when she walks at s km/h, the walk takes her 4 hours, including t minutes. So total time is 4 hours, which is 240 minutes. So walking time plus t minutes equals 240 minutes. So if walking time is 3.6 hours, which is 216 minutes, then t=24 minutes, which adds up to 240. Correct.\n\nSimilarly, second scenario: 2 hours 24 minutes is 144 minutes. Walking time at 4.5 km/h is 2 hours, which is 120 minutes, plus 24 minutes gives 144. Correct.\n\nTherefore, when walking at 3 km/h, walking time is 3 hours (180 minutes), plus 24 minutes, totaling 204 minutes. Therefore, the answer is 204 minutes.\n\nBut wait, the problem says \"the number of minutes\", so 204 minutes is the answer. But let me check if I did everything correctly. Let me re-calculate the quadratic solution.\n\nOriginal equations:\n\nEquation 1: 9/s + t/60 =4\n\nEquation 2: 9/(s+2) + t/60 =2.4\n\nSubtracting equation 2 from equation 1:\n\n9/s - 9/(s+2)=1.6\n\nWhich led to:\n\n18/(s(s+2))=1.6\n\nMultiply both sides by s(s+2):\n\n18=1.6*s*(s+2)\n\nDivide both sides by 1.6:\n\ns(s+2)=18/1.6=11.25\n\nThus, s² +2s -11.25=0\n\nMultiply by 4:\n\n4s² +8s -45=0\n\nQuadratic formula:\n\ns = [-8 ± sqrt(64 + 720)] /8 = [-8 ± sqrt(784)] /8 = [-8 ±28]/8\n\nPositive solution: (20)/8=2.5. Correct.\n\nTherefore, all steps are correct. Thus, the answer is 204 minutes.\n\nWait, but the problem says \"put your final answer within \\boxed{}\". Since the question asks for the number of minutes, so boxed 204.\n\n<omitted>

---

Token:1612 — **LIMO-P**

Okay, let's see. Aya goes for a 9-kilometer walk every morning and then stops at a coffee shop. The problem is about figuring out how long her entire outing takes when she walks at a speed of s plus a half kilometers per hour, including the time t spent in the coffee shop. Hmm. <Omitted> \n\nWait, so her original speed s is 2.5 km/h? That seems slow for walking, but maybe it's okay. Let's check.\n\nIf s is 2.5 km/h, then walking 9 km would take 9 / 2.5 = 3.6 hours, which is 3 hours and 36 minutes. Then, adding t minutes, the total time is 4 hours. So 3 hours 36 minutes plus t minutes equals 4 hours. Therefore, t must be 24 minutes.\n\nSimilarly, when she walks at s + 2 = 4.5 km/h, the walking time is 9 / 4.5 = 2 hours. Adding t = 24 minutes gives 2 hours + 24 minutes = 2 hours 24 minutes, which matches the second scenario. Okay, so that checks out.\n\nNow, the question is asking, if she walks at s + 1/2 km/h, so that's 2.5 + 0.5 = 3 km/h, how many minutes does the entire outing take, including t minutes in the coffee shop?\n\nSo, walking at 3 km/h, the walking time T_w is 9 / 3 = 3 hours. Converting that to minutes, 3 hours is 180 minutes. Adding t, which is 24 minutes, so total time is 180 + 24 = 204 minutes.\n\nWait, but let me confirm again. The problem says \"the walk takes her 4 hours, including t minutes...\" So the total time is 4 hours, which is 240 minutes. If walking time is 3 hours 36 minutes (which is 216 minutes) plus t minutes equals 240, then t is indeed 24 minutes. Similarly, in the second case, walking time is 2 hours (120 minutes) plus 24 minutes is 144 minutes, which is 2 hours 24 minutes. So that's consistent.\n\nTherefore, when walking at 3 km/h, walking time is 3 hours (180 minutes) plus 24 minutes in the coffee shop, totaling 204 minutes. So the answer should be 204 minutes.\n\n\n\nBut just to double-check, let me go through the steps again.\n\nOriginal speed s = 2.5 km/h, so walking 9 km takes 9/2.5 = 3.6 hours. 0.6 hours is 36 minutes, so 3 hours 36 minutes. Adding t minutes gives 4 hours. So 3h36m + t = 4h => t = 4h – 3h36m = 24m. Correct.\n\nAt s + 2 = 4.5 km/h, walking time is 2 hours. 2h + 24m = 2h24m. Correct.\n\nNow, s + 1/2 = 2.5 + 0.5 = 3 km/h. Walking time is 9/3 = 3 hours. Total time is 3h + 24m = 3h24m, which is 3*60 + 24 = 180 + 24 = 204 minutes. \n\nYes, that seems right. Therefore, the number of minutes is 204.\n\n**Final Answer**\nThe number of minutes the walk takes her, including the \\( t \\) minutes spent in the coffee shop, is \\\boxed{204}."

---

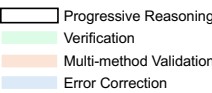

- ▢ Progressive Reasoning
- ▢ Verification
- ▢ Multi-method Validation
- ▢ Error Correction

Figure 9: Comparison of reasoning chains between model LIMO (left, 3,234 tokens) and PIR-optimized LIMO-P (right, 1,612 tokens) for the same mathematical problem. The model trained with PIR-optimized dataset maintains essential progressive reasoning while eliminating redundant verification steps, resulting in a 50% reduction in token count without sacrificing solution accuracy.

