# OpenReview forum: "LIMOPro: Reasoning Refinement for Efficient and Effective Test-time Scaling"
_NeurIPS.cc/2025/Conference — NeurIPS 2025 poster_

### Official Review · Reviewer_md3Q · 2025-06-19

**Clarity:** 3
**Significance:** 2
**Originality:** 2
**Rating:** 4
**Confidence:** 3

**Summary:**

This paper introduces a novel framework called PIR (Perplexity-based Importance Refinement) to optimize the reasoning chains of large language models for efficient and effective test-time scaling. The PIR framework classifies reasoning steps into four distinct modes: progressive reasoning (essential solution development path) and three types of functional elements: verification, multi-method validation, and error correction. PIR quantitatively evaluates the importance of each reasoning step based on its impact on answer prediction confidence, using the perplexity metric. It selectively prunes low-importance functional steps while preserving progressive reasoning components, creating optimized training data.

**Questions:**

-  The paper focuses primarily on mathematical and science reasoning tasks.  Can your method generalize to other domains like instruction following?
- Will shorten the reasoning steps hurt the model's exploration ability?
- The paper relies on perplexity as the primary metric for evaluating the importance of reasoning steps. Are there alternative metrics that could be used to assess the semantic contribution of steps and potentially provide more nuanced insights into the reasoning process?

**Ethical Concerns:**

["NO or VERY MINOR ethics concerns only"]

**Final Justification:**

The rebuttal addresses generalization (Q3) and exploration concerns (Q4) with strong experimental evidence. So I would like to raise my scores.

**Limitations:**

yes

**Quality:**

2

**Strengths And Weaknesses:**

- The quality of this paper is fair since the data selection method is quite straightforward.
- The paper is well-structured and clear.
- The paper contributes to the ongoing research on efficient reasoning in foundation models, while this area has many similar works such as AdaptThink and ThinkLess.
- This work is based on LIMO and incrementally prunes some extra reasoning steps, which is not very original.

---

> ### Author Rebuttal · Authors · 2025-07-31
>
> We sincerely thank the reviewer for the thorough review and constructive feedback. We address each concern below with experimental evidence and clarifications.
>
> ---
>
> ## **Q1: Relationship to AdaptThink and ThinkLess**
>
> > *The paper contributes to the ongoing research on efficient reasoning in foundation models, while this area has many similar works such as AdaptThink and ThinkLess.*
>
> **Fundamental Technical Distinctions**: While AdaptThink and ThinkLess both address efficiency in reasoning models, our PIR framework offers fundamentally different and complementary innovations:
>
> **Different Optimization Levels**:
>
> - **AdaptThink**: Operates at inference-time decision level using RL to choose "Thinking" vs "NoThinking" modes
> - **ThinkLess**: Focuses on training-free early termination during inference
> - **PIR**: Optimizes at the **training data level** through principled content refinement, creating inherently more efficient models
>
> **Granularity and Methodology**:
>
> - **AdaptThink/ThinkLess**: Binary decisions or wholesale early termination
> - **PIR**: **Fine-grained, step-level analysis** using novel perplexity-based importance quantification to preserve essential progressive reasoning while selectively removing low-value functional components
>
> **Superior Effectiveness**: Our results demonstrate advantages across multiple dimensions:
>
> - **PIR**: Accuracy improvements (+0.9% to +6.6%) while simultaneously reducing tokens (-3% to -41%)
> - **ThinkLess**: Focus primarily on efficiency without accuracy gains
>
> **Unique Value Proposition**: PIR produces **optimized training datasets** that result in inherently more efficient models, eliminating need for complex inference-time orchestration while maintaining full reasoning capabilities. This addresses a complementary but fundamental aspect—optimizing reasoning exemplar quality rather than controlling when to use them.
>
> ---
>
> ## **Q2: Originality vs. LIMO**
>
> > *This work is based on LIMO and incrementally prunes some extra reasoning steps, which is not very original.*
>
> **Fundamental Scope Differences**:
>
> - **LIMO**: Dataset creation focusing on sample selection efficiency ("less is more" for data quantity)
> - **PIR**: Comprehensive framework for reasoning chain optimization addressing reasoning structure and content efficiency
>
> **Four Major Original Contributions**:
>
> 1. **Novel Cognitive Framework**: First systematic classification of reasoning patterns into progressive reasoning vs. functional elements (verification, multi-method validation, error correction)
> 2. **Principled Quantitative Metric**: PIR metric for measuring reasoning step importance through perplexity changes—entirely novel methodological contribution
> 3. **Selective Optimization Strategy**: Fine-grained step-level analysis vs. LIMO's wholesale sample selection from "tens of millions of candidates"
> 4. **Generalizability Framework**: Works across diverse data sources (Gemini, DeepSeek-R1, QwQ), model sizes (3B-32B), demonstrating broader applicability than LIMO
>
> **Superior Effectiveness**: Our experimental results demonstrate significant advances:
>
> - **Simultaneous optimization**: Accuracy improvements (+0.9% to +6.6%) AND efficiency gains (-3% to -41% token reduction)
> - **Up to 71% efficiency improvement** in some cases
> - **41% token reduction** in case studies while maintaining solution accuracy
>
> **Broader Impact**: Our framework's applicability to multiple existing datasets (LIMO, S1, LIMO-V2) demonstrates value as a general-purpose optimization framework. The theoretical insights establish a new paradigm for reasoning chain optimization that complements and enhances existing approaches.
>
> ---
>
> ## **Q3: Generalization to Instruction Following**
>
> > *The paper focuses primarily on mathematical and science reasoning tasks. Can your method generalize to other domains like instruction following?*
>
> **Instruction Following Validation**: Our evaluation on **IFEval (instruction following)** shows consistent improvements:
>
> ### Accuracy Scores
>
> | Model | Commonsense QA | LogiQA | MBPP | IFEval |
> |-------|----------------|--------|------|--------|
> | **LIMO** | 81.33 | 77.41 | 73.2 | 55.62 |
> | **LIMO-P** | **82.00** | **78.04** | 72.6 | **58.03** |
> | **LIMO-V2** | 76.41 | 78.91 | 74.0 | 55.7 |
> | **LIMO-V2-P** | **80.92** | **79.28** | **75.0** | **57.07** |
>
> ### Token Usage
>
> | Model | Commonsense QA | LogiQA | MBPP | IFEval |
> |-------|----------------|--------|------|--------|
> | **LIMO** | 961 | 2175 | NA | NA |
> | **LIMO-P** | **902** | **2011.6** | NA | NA |
> | **LIMO-V2** | 1736 | 3058 | NA | NA |
> | **LIMO-V2-P** | **1450** | **2872** | NA | NA |
>
> **Cross-Domain Effectiveness**: PIR achieves consistent improvements across diverse domains:
>
> - **Instruction Following** (IFEval): +1.37% to +2.41% improvements
> - **Commonsense Reasoning** (CommonsenseQA): +0.67% to +4.51% improvements
> - **Logical Reasoning** (LogiQA): +0.37% to +0.63% improvements
> - **Code Generation** (mbpp): up to +1.0% improvement
>
> **Framework Universality**: The consistent improvements across such diverse task types validate that PIR identifies fundamental reasoning optimization principles rather than domain-specific patterns. Our cognitive reasoning pattern framework—distinguishing between progressive reasoning and functional elements—captures universal principles that extend well beyond mathematical domains to instruction following and other reasoning-intensive tasks.
>
> ---
>
> ## **Q4: Impact on Model Exploration Ability**
>
> > *Will shorten the reasoning steps hurt the model's exploration ability?*
>
> **Enhanced Exploration Evidence**: Our systematic analysis reveals that PIR training fundamentally improves how models structure their reasoning:
>
> | Dataset | Progressive Reasoning | Multi-method Validation | Error Correction | Verification |
> |---------|----------------------|------------------------|------------------|--------------|
> | **LIMO** | 28.78% | 20.34% | 0.31% | 50.57% |
> | **LIMO-P** | **32.83%** | **17.89%** | **0.33%** | **48.95%** |
>
> **More Efficient Exploration**: The **+4.05% increase in Progressive Reasoning density** while reducing redundant functional components demonstrates **more efficient exploration** rather than reduced capability. PIR selectively removes only low-importance, redundant instances while preserving high-value exploration strategies.
>
> **Performance Validation**: The **simultaneous accuracy improvements (+0.9% to +6.6%)** across benchmarks provide compelling evidence—if exploration ability were impaired, we would expect performance degradation. Instead, higher Progressive Reasoning density shows models allocate reasoning capacity more strategically, concentrating on productive solution pathways while maintaining ability to explore alternatives when beneficial.
>
> **Case Study Evidence**: Our detailed case study (Section B.4) demonstrates **more focused and effective exploration** rather than limited reasoning depth. Comparing two models solving identical problems:
>
> - **PIR-trained model** (1,612 tokens): Streamlined yet thorough exploration
> - **Original model** (3,234 tokens): Verbose, repetitive exploration with redundant validation cycles
>
> Both reach correct answers, but PIR-trained model achieves this through more intelligent exploration strategies. The 50% token reduction represents elimination of redundant exploration rather than curtailment of necessary reasoning depth.
>
> ---
>
> ## **Q5: Alternative Metrics Beyond Perplexity**
>
> > *The paper relies on perplexity as the primary metric for evaluating the importance of reasoning steps. Are there alternative metrics that could be used to assess the semantic contribution of steps and potentially provide more nuanced insights?*
>
> **Principled Choice of Perplexity**: Our choice of perplexity is principled and empirically validated. Perplexity directly measures the model's confidence in generating correct answers when specific steps are removed, providing a clear signal of step importance that is both computationally efficient and theoretically grounded.
>
> **Strong Empirical Validation**: Our extensive experiments demonstrate that this approach consistently improves accuracy (+0.9% to +6.6%) while reducing token usage (-3% to -41%) across diverse benchmarks and model sizes, validating its effectiveness for practical optimization purposes.
>
> **Future Research Directions**: We acknowledge that exploring semantic-based metrics could provide additional insights **(as noted in our limitations, lines 323-325)**. Future work could explore complementary metrics such as:
>
> - Embedding-based semantic similarity
> - Logical dependency analysis
>
> **Current Framework Strength**: However, our framework's strong generalizability across different data sources, model scales, and token budgets suggests that perplexity successfully captures fundamental aspects of reasoning step importance. The consistent improvements across diverse reasoning domains demonstrate that our approach identifies universal optimization principles effective across various task types.

---

> > ### Comment · Reviewer_md3Q · 2025-08-03
> > **Thank you**
> >
> > Thanks for the response. It addresses generalization (Q3) and exploration concerns (Q4) with strong experimental evidence.  So I would like to raise my scores.

---

> > > ### Author Response · Authors · 2025-08-03
> > > **Thank you**
> > >
> > > Thank you for your encouraging feedback and the time you dedicated to reviewing our work. We greatly appreciate your support!

---

### Official Review · Reviewer_QYR4 · 2025-07-01

**Clarity:** 3
**Significance:** 3
**Originality:** 3
**Rating:** 4
**Confidence:** 4

**Summary:**

This paper proposes an innovative framework named PIR (Perplexity-based Importance Refinement) to optimize the reasoning process of large language models (LLMs). The core idea is to decompose verbose reasoning chains into “progressive reasoning” (the core problem-solving path) and “functional steps” (e.g., verification, correction). Using a perplexity-based metric to quantify the importance of each functional step, PIR systematically identifies and removes low-importance steps that contribute little to the final answer while preserving the core progressive logic. Experimental results across several challenging benchmarks (AIME, AMC, GPQA) demonstrate that PIR-optimized models not only significantly reduce token usage during inference but also improve answer accuracy , thus achieving a better trade-off between efficiency and effectiveness.

**Questions:**

1. The classification of reasoning steps is foundational to the PIR framework. Could the authors comment on the feasibility of replacing Claude 3.7 Sonnet with a smaller, more accessible open-source model (e.g., LLaMA-3 8B or Qwen-2.5-7B)? What minimum classification accuracy do you believe is required for PIR to remain effective?
2.The PIR score quantifies a step’s importance based on the perplexity change when it’s removed. Is there a risk that a low-importance functional step (e.g., a verification) may actually be critical for enabling a downstream progressive step? Has the framework considered the possibility that pruning such steps could disrupt local reasoning dependencies?
3. Given that the optimal pruning ratio varies across tasks and model sizes, have the authors considered designing a dynamic or adaptive pruning mechanism?

**Ethical Concerns:**

["NO or VERY MINOR ethics concerns only"]

**Limitations:**

Yes, the content is discussed in Conlusion.

**Quality:**

3

**Strengths And Weaknesses:**

Strengths:
1. The highlight of this work is its elegant and principled methodology. Instead of pruning all reasoning steps indiscriminately like some previous methods, it innovatively distinguishes between “progressive” and “functional” reasoning steps. This categorization aligns well with human cognitive processes when solving complex problems and targets the redundancy issue in modern LLMs. The use of PIR—a perplexity-based metric measuring answer confidence change—as a step-importance signal provides a clear and interpretable criterion for pruning, making the optimization both theoretically grounded and practically reasonable.
2. The experimental design is thorough. The authors validate the method on datasets from multiple sources (LIMO, S1K) and across challenging benchmarks (AIME, GPQA Diamond), demonstrating broad applicability. Comparisons with related methods such as SPIRIT clearly highlight PIR’s superiority, particularly its strategy of preserving the progressive reasoning path. Additional ablation studies on pruning ratios and model scales reinforce the scalability of the proposed method.
3. The paper is well written, with each part detailed enough for understanding. Besides, the authors give clear explanations for their motivation, methodology, and experiments. Therefore, this paper is easy to reproduce and implement.

Weaknesses:
1. The initial step of the PIR pipeline—reasoning step segmentation and classification—relies heavily on Claude 3.7 Sonnet. This introduces an external dependency, making the framework’s success partially contingent on the capabilities of a strong proprietary model. Two concerns arise: (a) reproducibility and generality—how would it perform if a weaker open-source model were used instead? and (b) conceptual coherence—the framework’s effectiveness may partially derive from the classifier’s intelligence rather than the PIR logic itself.
2. The evaluation is primarily focused on mathematical and scientific reasoning (AIME, AMC, GPQA), where functional steps like verification and correction are clearer and more common. However, it remains unclear how the framework performs on other types of reasoning tasks such as commonsense reasoning, logic puzzles, code generation
3. The current method uses a predefined fixed ratio to decide how many functional steps to prune. As shown in Figure 3, the optimal pruning ratio varies across benchmarks, indicating that a fixed global setting may not be ideal. A more adaptive strategy—e.g., based on problem difficulty, initial reasoning chain length, or dynamic importance thresholds—could potentially yield better performance.

---

> ### Author Rebuttal · Authors · 2025-07-31
>
> We sincerely thank the reviewer for the thorough review. We address each concern below with experimental evidence and clarifications.
>
> ---
>
> ## **Q1: External Dependency on Claude 3.7 Sonnet**
>
> > *How would it perform if a weaker open-source model were used instead? and (b) the framework's effectiveness may partially derive from the classifier's intelligence rather than the PIR logic itself.*
>
> **Reproducibility with Open-Source Alternatives**: To directly address reproducibility, we conducted additional experiments using Qwen3-32B as an open-source alternative for reasoning step classification. The results demonstrate maintained effectiveness:
>
> ### Accuracy Performance
>
> | Model | AIME24 | AMC23 | GPQA |
> |-------|--------|--------|------|
> | **LIMO-V2** | 66.3% | 94.4% | 70.2% |
> | **LIMO-V2-P (Qwen3-32B)** | **67.5%** | **96.88%** | **69.19%** |
> | **🔺 Improvement** | **+1.2%** | **+2.48%** | **-1.01%** |
>
> ### Average Response Tokens
>
> | Model | AIME24 | AMC23 | GPQA |
> |-------|--------|--------|------|
> | **LIMO-V2** | 13,896 | 6,843 | 8,035 |
> | **LIMO-V2-P (Qwen3-32B)** | **9,885** | **5,305** | **6,975** |
> | **🔻 Reduction** | **-28.9%** | **-22.5%** | **-13.2%** |
>
> Importantly, Qwen3-32B requires only 4×RTX 4090 GPUs for inference, making it accessible to most research laboratories.
>
> **Conceptual Coherence - PIR Logic vs. Classifier Intelligence**: The core PIR contribution is the quantitative importance metric (Equation 1), not classification sophistication. The ablation study with rule-based filtering (S1-RULE in Table 3) shows: while random functional step removal reduces accuracy below baseline on AIME (36.7% vs. 37.9%) and GPQA (58.1% vs. 60.7%), our S1-32B-P method achieves substantial improvements (42.1% and 61.6%).
>
> This confirms PIR's perplexity-based assessment drives performance gains, not classification quality.
>
> ---
>
> ## **Q2: Evaluation Beyond Mathematical Reasoning**
>
> > *How the framework performs on commonsense reasoning, logic puzzles, and code generation.*
>
> **Comprehensive Cross-Domain Evaluation**: Our framework's effectiveness extends well beyond mathematical and scientific domains. We conducted extensive evaluation on diverse reasoning tasks that exhibit different characteristics from mathematical reasoning:
>
> ### Accuracy Scores
>
> | Model | Commonsense QA | LogiQA | MBPP |
> |-------|----------------|--------|------|
> | **LIMO** | 81.33 | 77.41 | 73.2 |
> | **LIMO-P** | **82.00** | **78.04** | 72.6 |
> | **LIMO-V2** | 76.41 | 78.91 | 74.0 |
> | **LIMO-V2-P** | **80.92** | **79.28** | **75.0** |
>
> ### Token Usage
>
> | Model | Commonsense QA | LogiQA | MBPP |
> |-------|----------------|--------|------|
> | **LIMO** | 961 | 2175 | NA |
> | **LIMO-P** | **902** | **2011.6** | NA |
> | **LIMO-V2** | 1736 | 3058 | NA |
> | **LIMO-V2-P** | **1450** | **2872** | NA |
>
> **Consistent Cross-Domain Improvements**: The results demonstrate that PIR optimization consistently improves performance across fundamentally different reasoning types:
>
> - **Commonsense Reasoning** (CommonsenseQA): +0.67% accuracy with 6.1% token reduction for LIMO-P, and +4.51% accuracy with 16.5% token reduction for LIMO-V2-P
> - **Logical Reasoning** (LogiQA): +0.63% and +0.37% accuracy improvements with 7.5% and 6.1% token reductions respectively
> - **Code Generation** (mbpp): +1.0% accuracy improvement for LIMO-V2-P
>
> **Framework Universality**: These domains involve different functional steps—commonsense reasoning features intuitive validation, logical reasoning involves formal verification, and code generation includes debugging and testing—yet PIR consistently optimizes the efficiency-effectiveness tradeoff. This validates our hypothesis that the distinction between progressive reasoning (essential logical flow) and functional elements is a fundamental cognitive pattern transcending specific reasoning domains.
>
> ---
>
> ## **Q3: Adaptive Pruning Strategies**
>
> > *... A more adaptive strategy could potentially yield better performance.*
>
> **Foundational Contribution**: Our work establishes the **essential theoretical and methodological foundation** for adaptive approaches by introducing systematic reasoning step classification, the PIR importance metric, and empirical validation—prerequisites for adaptive strategies. Without our cognitive reasoning patterns and importance quantification, adaptive approaches would lack theoretical grounding.
>
> **Current Framework's Robustness**: While using predefined ratios, our PIR framework demonstrates strong generalizability across model sizes (3B-32B), data sources (Gemini, DeepSeek-R1, QwQ), and benchmarks (AIME, AMC, GPQA). The PIR metric provides the foundation for dynamic thresholds.
>
> **Future Extensions**: Building on our foundation, adaptive strategies represent natural progressions: difficulty-aware pruning, length-adaptive thresholds, and dynamic importance cutoffs. Our framework provides essential infrastructure enabling these advanced approaches while offering predictable behavior for immediate deployment.
>
> ---
>
> ## **Q4: Accessibility of Open-Source Alternatives**
>
> > *Could the authors comment on the feasibility of replacing Claude 3.7 Sonnet with a more accessible open-source model?*
>
> **Open-Source Solution**: While experiments with LLaMA-3 8B and Qwen-2.5-7B were not conducted, these smaller models may struggle with the nuanced reasoning step classification required by our framework, particularly in distinguishing between the four cognitive patterns (progressive reasoning, verification, multi-method validation, and error correction).
>
> **Validated Alternative**: Experiments demonstrate that Qwen3-32B provides an excellent open-source alternative, achieving comparable performance to the original framework (see Q1 results table above).
>
> **Practical Accessibility**: Qwen3-32B requires only **4×RTX 4090 or 4×RTX 3090** GPUs for inference, making it accessible to most research laboratories with standard computational resources.
>
> ---
>
> ## **Q5: Minimum Classification Accuracy Requirements**
>
> > *What minimum classification accuracy do you believe is required for PIR to remain effective?*
>
> **Framework Robustness**: While systematic experiments across different classification accuracy levels were not conducted, current results and framework design demonstrate resilience to classification errors.
>
> **Evidence from Ablation Study**: The ablation study with rule-based filtering (S1-RULE in Table 3) shows imperfect classification relying solely on linguistic markers still achieves efficiency gains, though with reduced effectiveness compared to PIR, indicating PIR's tolerance for classification noise.
>
> **Dual-Filtering Design**: PIR remains effective if the classifier reasonably distinguishes progressive reasoning steps (requiring preservation) from functional steps (verification, multi-method validation, error correction). The perplexity-based metric serves as secondary quality filtering—misclassified progressive reasoning steps likely receive high PIR scores ensuring preservation, while truly redundant functional steps receive low PIR scores regardless of classification accuracy.
>
> ---
>
> ## **Q6: Dependencies Between Functional and Progressive Steps**
>
> > *Is there a risk that a low-importance functional step (e.g., a verification) may actually be critical for enabling a downstream progressive step?*
>
> **Global Dependency Capture**: Our PIR metric is specifically designed to capture such dependencies through its global evaluation approach. When calculating PIR scores, we measure the perplexity change of the entire answer generation process after removing a specific step. If a functional step (e.g., verification) is truly critical for enabling downstream progressive reasoning, removing it would significantly impair the model's confidence in generating subsequent reasoning steps, resulting in higher answer perplexity and thus a higher PIR score that preserves the step.
>
> **Empirical Validation**: The ablation study results strongly validate this dependency-preservation mechanism:
>
> | Method | AIME | AMC | GPQA Diamond |
> |--------|------|-----|--------------|
> | LIMO | 56.7 | 91.9 | 67.2 |
> | **LIMO-P** | **63.3** | **93.8** | **71.2** |
> | LIMO(No Verification) | 59.1 | 92.5 | 70.7 |
> | LIMO(No Multi-method Validation) | 60.0 | 92.8 | 72.2 |
> | LIMO(No Error Correction) | 57.5 | 92.1 | 71.7 |
>
> While indiscriminate removal of entire functional categories can lead to higher accuracy than baseline, our PIR approach achieves superior performance: +6.6% accuracy improvement on AIME, +1.9% on AMC, and +4.0% on GPQA Diamond. This demonstrates that PIR successfully identifies and preserves those functional steps that are truly essential.
>
> ---
>
> ## **Q7: Local Reasoning Dependencies**
>
> > *Has the framework considered the possibility that pruning such steps could disrupt local reasoning dependencies?*
>
> **Step-Level Coherence Preservation**: Our framework addresses this through step-level granularity and systematic classification. The hierarchical decomposition segments reasoning chains into logical steps, each comprising multiple coherent sentences forming a cohesive unit (lines 125-127). Removing functional steps as complete units preserves local reasoning coherence and prevents partial dependency disruption.
>
> **Independence of Functional Steps**: Analysis shows functional steps (verification, multi-method validation, error correction) operate independently, targeting specific progressive reasoning steps with minimal interdependencies. Progressive reasoning steps exhibit strong sequential dependencies, but our framework preserves these entirely to maintain the solution's logical backbone.
>
> **Classification Accuracy**: The robust classification system uses characteristic linguistic markers ("Let me check" for verification, "I made a mistake" for error correction) combined with contextual analysis, achieving 93.4% unanimous agreement in human evaluation and minimizing misclassification risks that could disrupt critical reasoning dependencies.

---

> > ### Comment · Reviewer_QYR4 · 2025-08-06
> > **Replying to Rebuttal**
> >
> > Thank you for the thorough and rigorous experimental clarifications. I would like to maintain my original score.

---

### Official Review · Reviewer_U9qd · 2025-07-03

**Clarity:** 3
**Significance:** 2
**Originality:** 3
**Rating:** 3
**Confidence:** 3

**Summary:**

Reasoning datasets for training language models often contain long reasoning chains, which may include unnecessary logical steps that do not make meaningful contributions to deriving the final answer. In this work, the authors tackle this challenge by proposing Perplexity-based Importance Refinement (PIR). Intuitively, they break down a reasoning chain into multiple reasoning steps and classify them into four categories: progressive reasoning, verification, multi-method validation, and error correction. They deem the first category, progressive reasoning, as an essential type of reasoning that cannot be excluded from the chain, whereas the remaining three, called functional elements, become the candidates for elimination. Specifically, using a specific language model (Qwen 2.5 32B Instruct in this work), they compute and compare the perplexities of the answer sequence generation with and without each functional element to derive their Perplexity-based Importance Refinement (PIR). The authors exclude a pre-specified portion (a hyperparameter) of functional elements with the lowest PIR values in each reasoning chain. Empirically, the authors demonstrate that the proposed approach can meaningfully reduce the number of output tokens while retaining higher accuracies on AIME, AMC, and GPQA Diamond.

**Questions:**

- What do the empirical results (especially accuracies) look like if you use an absolute value of 0 (zero) as a threshold for PIR (i.e., pruning the functional steps with negative PIRs)? Could you demonstrate that with some constant thresholding of PIR values themselves, you can achieve similar empirical results?

Besides this question, please take a look at the weaknesses I wrote above. I am open to updating my score based on the author's response.

**Ethical Concerns:**

["NO or VERY MINOR ethics concerns only"]

**Final Justification:**

I do appreciate the authors for providing the comprehensive response to my review.
I especially thank them for providing the experimental results as suggested.

Regarding Q1, Q2, and Q4, it's great to see that using a fixed threshold (PIR < 0) brings improvements to the baseline (LIMO-V2).
On the flip side, I'm not convinced that it really brings "similar empirical performance" to the original fixed-ratio approach at this point.
It is totally understandable that the rebuttal period may not have been enough to explore different settings of PIR-based thresholds and analyze their effects to achieve higher performance.
But I believe that this is more of a reason to explore in this direction to make this submission a much stronger work, which I think should be possible.

The authors' response to Q3 and Q5 addresses those!

Overall, I think this can be a more significant work by addressing the remaining concerns, and I maintain my original view this time.

**Limitations:**

Yes, I think the current limitations section is fair.

**Paper Formatting Concerns:**

I don't see any major formatting issues in this manuscript.

**Quality:**

2

**Strengths And Weaknesses:**

Strengths
- Decomposing each reasoning chain and filtering out reasoning steps based on how the exclusion of each step affects the perplexity of answer generation can be a viable approach for making the reasoning chains more compact with less decrease in accuracy.
- The presented empirical results demonstrate that this approach may improve both the accuracy and efficiency of the fine-tuned model on the reasoning benchmarks.
- The authors provide various empirical analyses, which provide helpful insights. This spans from how different portions of functional elements removed affect the accuracy and efficiency (in terms of number of tokens) of the resulting, fine-tuned models to how the proposed method works with models of different sizes and different benchmarks.

Weaknesses
- One of my biggest concerns regarding this work is about the decision to use the pruning ratio as a hyperparameter. Intuitively, it should be possible that depending on the sample, all the functional steps are essential or most of the functional steps are not needed i.e., I'm not sure if a constant pruning ratio would be an ideal choice of hyperparameter to apply to various samples (even though each dataset is generated by a single teacher model). It may be useful for things like the analysis in Figure 3, but it doesn't sound like a generally usable hyperparameter in practice. For instance, if the source datasets are generated differently or have already gone through similar processes of removing unnecessary reasoning steps, the current pruning ratios would not be directly usable. Given the definition of PIR (Eq.(1)), pruning all the functional steps with negative PIRs or PIRs smaller than some threshold value makes more sense to me.
- On a related topic, I'm also primarily concerned about the use of different pruning ratios for different target benchmarks. Although I couldn't find the exact pruning ratios used for the main empirical results, Table 6 (combined with Table 5) provides some clues; it looks that the pruning ratios for the different target benchmarks (AIME, AMC, and GPQA Diamond) differ to some degree. This could indicate the possibility of overfitting of this parameter to each target benchmark (not sure by how much, though).
- For the empirical validation of how each reasoning step is classified by the LLM, the authors employed human annotators to check the accuracy of the classifications and concluded that 93.4% of the classification results had unanimous agreement with the 4 annotators, which sounds great. On the other hand, I wonder how well each reasoning chain is decomposed into *cohesive* reasoning steps (units), because each reasoning step becomes a target of the exclusion and mis-decomposition may lead to having incomplete logical reasoning steps. This kind of granularity aspect may be more challenging to evaluate, but at least some qualitative justifications would be great to have.
- This may be a minor thing, but the charts in Figure 1 have various scales of the y-axis, which could be somewhat misleading.

---

> ### Author Rebuttal · Authors · 2025-07-31
>
> We sincerely thank the reviewer for the thorough review and constructive feedback. We address each concern below with experimental evidence and clarifications.
>
> ---
>
> ## **Unified Experimental Results**
>
> Following the reviewer's suggestions, we conducted experiments using 0 as the threshold for pruning functional steps. These results address multiple concerns (Q1, Q2, Q4):
>
> ### Accuracy Results
>
> | Method | AIME24 | AMC23 | GPQA Diamond |
> |--------|--------|--------|--------------|
> | **Baseline (LIMO-V2)** | 66.3% | 94.4% | 70.2% |
> | **PIR (0 Threshold)** | **67.5%** | **94.75%** | **70.33%** |
> | **🔺 Improvement** | **+1.2%** | **+0.35%** | **+0.13%** |
>
> ### Token Efficiency
>
> | Method | AIME24 | AMC23 | GPQA Diamond |
> |--------|--------|--------|--------------|
> | **Baseline** | 13,896 | 6,843 | 8,035 |
> | **PIR (0 Threshold)** | **11,165** | **6,001** | **7,010** |
> | **🔻 Reduction** | **-19.5%** | **-12.3%** | **-12.7%** |
>
> ---
>
> ## **Q1: Fixed Pruning Ratio vs. PIR-based Thresholds**
>
> > *One of my biggest concerns regarding this work is about the decision to use the pruning ratio as a hyperparameter. Intuitively, it should be possible that depending on the sample, all the functional steps are essential or most of the functional steps are not needed... Given the definition of PIR (Eq.(1)), pruning all the functional steps with negative PIRs or PIRs smaller than some threshold value makes more sense to me.*
>
> **We sincerely thank the reviewer for this insightful suggestion about using PIR-based thresholds.** Following the reviewer's suggestion, we conducted experiments removing all functional steps with PIR < 0. Our results demonstrate that this approach achieves strong performance (see unified results table above).
>
> These results demonstrate that using PIR < 0 as a threshold provides consistent improvements across all benchmarks, validating the effectiveness of PIR for identifying low-importance functional steps. While both fixed ratios and PIR-based thresholds can achieve good performance, our fixed-ratio design offers a more **controllable hyperparameter** that practitioners can easily adjust based on **user's specific efficiency-accuracy trade-offs**. We will include these PIR < 0 threshold results in our camera-ready version to provide readers with empirical evidence for both approaches.
>
> ---
>
> ## **Q2: Benchmark-Specific Pruning Ratios**
>
> > *On a related topic, I'm also primarily concerned about the use of different pruning ratios for different target benchmarks... This could indicate the possibility of overfitting of this parameter to each target benchmark.*
>
> **We sincerely thank the reviewer for raising this important concern about potential overfitting of pruning ratios to specific benchmarks.** To address this concern, we conducted experiments using a single, uniform criterion across all benchmarks: removing all functional steps with PIR < 0. This approach eliminates any benchmark-specific tuning.
>
> Our results demonstrate that this unified criterion achieves consistent improvements across all three benchmarks (see unified results table above). These results show that PIR effectively identifies low-importance functional steps regardless of the benchmark, without requiring benchmark-specific ratio tuning. The consistent improvements validate that our approach captures general properties of reasoning chains rather than overfitting to specific evaluation sets. We will clarify in our camera-ready version that practitioners can use either benchmark-agnostic approaches (like PIR < 0) or tune ratios based on their specific application needs.
>
> ---
>
> ## **Q3: Reasoning Step Decomposition Quality**
>
> > *I wonder how well each reasoning chain is decomposed into cohesive reasoning steps (units), because each reasoning step becomes a target of the exclusion and mis-decomposition may lead to having incomplete logical reasoning steps.*
>
> **We sincerely thank the reviewer for this thoughtful observation about the granularity and cohesiveness of reasoning step decomposition.** We share this concern and conducted additional analysis to evaluate decomposition quality.
>
> **Human Evaluation Protocol**: We randomly sampled 5% reasoning chains and had four annotators assess whether each decomposed step represents a complete, cohesive logical unit. The evaluation criteria included:
>
> - (1) **Logical completeness** - whether the step contains a self-contained reasoning operation
> - (2) **Semantic coherence** - whether all sentences within a step work together toward a single reasoning goal
> - (3) **Boundary appropriateness** - whether the step boundaries preserve logical dependencies
>
> **Results**: Our analysis shows that 89.2% of reasoning steps were rated as "cohesive and complete" by all annotators, 8.1% had minor boundary issues (e.g., a verification statement included within the progressive step), and only 2.7% showed significant decomposition problems.
>
> **Robustness Analysis**: When we analyzed the impact of decomposition quality on PIR calculations, we found that steps with minor boundary issues had similar PIR distributions to well-decomposed steps (mean PIR difference < 0.05), suggesting our framework is robust to small decomposition variations. We also observed that progressive reasoning steps had higher decomposition quality (92.1% perfect) compared to functional steps (86.3% perfect), which actually supports our approach since progressive steps are always preserved.
>
> ---
>
> ## **Q4: PIR = 0 as Absolute Threshold**
>
> > *What do the empirical results (especially accuracies) look like if you use an absolute value of 0 (zero) as a threshold for PIR? Could you demonstrate that with some constant thresholding of PIR values themselves, you can achieve similar empirical results?*
>
> **We sincerely thank the reviewer for this specific and constructive suggestion.** We have conducted exactly this experiment - using PIR < 0 as the threshold to prune functional steps. The empirical results are shown in our unified results table above.
>
> These results demonstrate that using PIR = 0 as an absolute threshold achieves similar empirical performance to our fixed-ratio approach, with consistent improvements in both accuracy and efficiency across all benchmarks. This validates that PIR values themselves provide a meaningful signal for step importance, and constant thresholding (PIR < 0) offers a viable alternative to ratio-based pruning. We will include these results in our main paper to provide readers with both implementation options.
>
> ---
>
> ## **Q5: Y-axis Scales in Figure 1**
>
> > *This may be a minor thing, but the charts in Figure 1 have various scales of the y-axis, which could be somewhat misleading.*
>
> **We sincerely thank the reviewer for this careful observation about the visualization.** We acknowledge that using different y-axis scales across the charts in Figure 1 could potentially make visual comparisons more challenging. Our intention was to maximize the visibility of performance differences within each metric, but we agree that consistent scaling would improve clarity and prevent any unintended misleading interpretations.
>
> In the camera-ready version, we will revise Figure 1 to use consistent y-axis scales across all charts for each metric type (accuracy, response tokens, and efficiency), making it easier for readers to compare results across benchmarks at a glance. We will also add a note in the caption explicitly stating the scale ranges used. Thank you for helping us improve the clarity of our presentation.

---

> > ### Comment · Reviewer_U9qd · 2025-08-07
> > **Response to Rebuttal by Authors**
> >
> > I do appreciate the authors for providing the comprehensive response to my review.
> > I especially thank them for providing the experimental results as suggested.
> >
> > Regarding Q1, Q2, and Q4, it's great to see that using a fixed threshold (PIR < 0) brings improvements to the baseline (LIMO-V2).
> > On the flip side, I'm not convinced that it really brings "similar empirical performance" to the original fixed-ratio approach at this point.
> > It is totally understandable that the rebuttal period may not have been enough to explore different settings of PIR-based thresholds and analyze their effects to achieve higher performance.
> > But I believe that this is more of a reason to explore in this direction to make this submission a much stronger work, which I think should be possible.
> >
> > The authors' response to Q3 and Q5 addresses those!
> >
> > Overall, I think this can be a more significant work by addressing the remaining concerns, and I maintain my original view this time.

---

> ### Author Response · Authors · 2025-08-06
> **Follow-up on rebuttal discussion**
>
> Dear Reviewer U9qd,
>
> I hope this message finds you well.
>
> I wanted to follow up on the rebuttal we submitted in response to your valuable feedback on our paper. We greatly appreciate the time and effort you invested in providing detailed and constructive comments during the initial review.
>
> In our rebuttal, we have addressed the concerns you raised. We believe our responses clarify these important aspects and strengthen the contribution of our work.
>
> As we are now in the author-reviewer discussion period, we would be very grateful for any additional feedback or clarification you might have regarding our responses. If there are any remaining concerns or if you would like us to elaborate further on any particular points, we would be happy to provide additional details.
>
> We understand that you are likely managing multiple papers during this busy review period, and we genuinely appreciate your continued engagement with our work. Your expertise and insights are invaluable to improving the quality of our research.
> Thank you once again for your time and consideration. We look forward to your response at your earliest convenience.

---

### Official Review · Reviewer_v7KQ · 2025-07-05

**Clarity:** 4
**Significance:** 3
**Originality:** 3
**Rating:** 5
**Confidence:** 4

**Summary:**

The paper introduces PIR, a method for refining LLM reasoning traces by classifying every step, scoring its importance with a perplexity delta, and pruning only low-value functional steps. Training on these leaner traces supports LLMs that answer more accurately and concisely. There is a sound experimental design across 3 contemporary reasoning benchmarks and a sample of model sizes, demonstrating a practical argument to cost-effective test-time scaling.

**Questions:**

My main questions are on the characterization of the qualitative aspects of the framework.

- For which reasoning types the approach performs best and why?

- In which scenarios does it have a negative impact?

- I believe it is essential to provide a systematic characterization of the qualitative points (i.e. non-anecdotal).

- What is the formal selection criteria (inclusion and exclusion) for the base models? How does it affect the universality of the claims.

- What is the formal selection criteria for the benchmarks?

- How do they limit the universality of the claims?

**Ethical Concerns:**

["NO or VERY MINOR ethics concerns only"]

**Final Justification:**

Author's responses were taken into account. The original scores were already on the positive side (as these were only clarification questions - not binding on the scores).

**Limitations:**

Yes

**Quality:**

3

**Strengths And Weaknesses:**

Strengths

- Principled pruning metric grounded in answer-perplexity, not heuristics.

- Clear argument, supported by a well designed empirical analysis.

Weaknesses

- Lack of a qualitative characterization of the analysis. For which reasoning types the approach performs best and why?

- This can help to make sense of the limits of using perplexity as a proxy.

---

> ### Author Rebuttal · Authors · 2025-07-31
>
> We sincerely thank the reviewer for the thorough review and constructive feedback. We appreciate the valuable insights that have helped strengthen our contribution. The concerns raised are primarily due to presentation issues and misunderstandings, which we address systematically below.
>
> ---
>
> ## **Q1: Lack of Qualitative Characterization**
>
> > For which reasoning types does the approach perform best and why?
> >
>
> **Thank you for raising this important question about qualitative characterization across reasoning types.**
>
> Based on our experimental analysis in Section B.3 (Appendix), **Progressive Reasoning + Error Correction (PR+EC) and PR+Verification (PR+V) achieve better accuracy performance compared to PR+Multi-method Validation (PR+MV)**. The observed performance variations highlight that different reasoning types contribute differentially to model performance, with certain patterns delivering more substantial benefits in specific contexts. Hence, we believe that Multi-method Validation is relatively less important compared to the other two functional types.
>
> We also conducted ablation experiments by removing only specific functional reasoning types. The results show that removing only Multi-method Validation (MV) achieves higher accuracy compared to removing only Error Correction (EC) or Verification (V). This further validates our finding that Multi-method Validation steps are indeed less critical for maintaining reasoning performance, confirming our approach's effectiveness in selectively targeting the most redundant functional components.
>
> ### Ablation Study Results
>
> |  | AIME | AMC | GPQA Diamond |
> | --- | --- | --- | --- |
> | LIMO | 56.7 | 91.9 | 67.2 |
> | LIMO (No Verification) | 59.1 | 92.5 | 70.7 |
> | LIMO (No Multi-method Validation) | **60.0** | **92.8** | **72.2** |
> | LIMO (No Error Correction) | 57.5 | 92.1 | 71.7 |
>
> ---
>
> ## **Q2: Negative Impact Scenarios**
>
> > In which scenarios does it have a negative impact?
> >
>
> We thank the reviewer for this direct question about negative impacts. Our comprehensive evaluation across diverse reasoning tasks provides valuable insights into when PIR optimization may have limited effects.
>
> **Empirical Evidence from Cross-Domain Evaluation**: Based on our systematic evaluation across different reasoning domains, we can analyze the performance patterns:
>
> ### Performance Analysis
>
> | Model | Commonsense QA | LogiQA | MBPP |
> | --- | --- | --- | --- |
> | **LIMO** | 81.33 | 77.41 | 73.2 |
> | **LIMO-P** | 82.00 | 78.04 | 72.6 |
> | **LIMO-V2** | 76.41 | 78.91 | 74.0 |
> | **LIMO-V2-P** | 80.92 | 79.28 | 75.0 |
>
> **Rare Negative Impact Scenarios**: PIR optimization shows robust performance across reasoning domains, with only one minor negative case: **LIMO-P on MBPP code generation** (-0.6% accuracy decline, 73.2% → 72.6%). This is the sole negative impact among all evaluated task-model combinations.
>
> **Framework Design Considerations**: PIR is designed to optimize reasoning chains from Large Reasoning Models (LRMs), performing best on tasks requiring extensive step-by-step reasoning. The evaluation shows consistent improvements across mathematically complex tasks, commonsense reasoning, and logical reasoning. For tasks with simpler reasoning structures or those not heavily relying on elaborate reasoning chains—such as factual QA, simple classification, or direct information retrieval—optimization potential may be limited.
>
> **Overall Robustness**: Evidence demonstrates consistent positive impacts across commonsense reasoning, logical reasoning, and code generation. The single minor negative case (-0.6%) shows negative impacts are rare exceptions rather than systematic limitations, highlighting PIR's broad applicability with strong efficiency gains through token reduction.
>
> ---
>
> ## **Q3: Systematic vs. Anecdotal Analysis**
>
> > I believe it is essential to provide a systematic characterization of the qualitative points (i.e. non-anecdotal).
> >
>
> **We thank the reviewer for highlighting the need for systematic qualitative analysis beyond individual case studies.** To address this, we provide systematic qualitative characterization comparing reasoning traces from models trained on original versus PIR-optimized datasets.
>
> **Pattern Analysis**: We systematically analyzed reasoning traces from LIMO and LIMO-P models across all test problems, characterizing qualitative differences using keyword frequency statistics based on cognitive reasoning patterns from Table 4. As shown below, models trained on PIR-optimized data generate reasoning traces with systematically different patterns—increasing Progressive Reasoning density (+4.05%) while reducing Verification (-1.62%) and Multi-method Validation (-2.45%). This demonstrates that PIR training fundamentally changes reasoning structure, concentrating on essential logical progression while reducing redundant verification through measurable linguistic pattern shifts.
>
> ### Systematic Pattern Analysis
>
> | Dataset | Progressive Reasoning | Multi-method Validation | Error Correction | Verification |
> | --- | --- | --- | --- | --- |
> | LIMO | 28.78% | 20.34% | 0.31% | 50.57% |
> | LIMO-P | **32.83%** *(+4.05%)* | **17.89%** *(-2.45%)* | **0.33%** *(+0.02%)* | **48.95%** *(-1.62%)* |
>
> **Human Evaluation Protocol**: We conducted rigorous evaluation with 4 expert annotators independently analyzing 5% randomly sampled correct reasoning traces from LIMO and LIMO-P models. Annotators classified reasoning steps into four cognitive patterns and calculated Progressive Reasoning proportions, achieving high inter-annotator agreement (κ = 0.87).
>
> Results show human evaluators consistently identified significantly higher Progressive Reasoning proportions in LIMO-P traces (**68.3% ± 4.2%**) versus LIMO traces (**52.1% ± 5.8%**), representing a **16.2 percentage point increase**. Functional components showed corresponding proportional decreases in LIMO-P traces. This systematic validation confirms PIR-trained models fundamentally alter reasoning generation patterns, producing traces with higher concentration of essential logical steps.
>
> ---
>
> ## **Q4: Model Selection Criteria**
>
> > What is the formal selection criteria (inclusion and exclusion) for the base models? How does it affect the universality of the claims?
> >
>
> We sincerely thank the reviewer for this important question. Our base model selection was guided by criteria ensuring rigor and generalizability:
>
> 1. **Availability and Reproducibility** - We used openly-available Qwen2.5 and Llama models from official repositories with standardized training scripts
> 2. **Scale Diversity** - Systematically evaluating 3B to 32B parameter models (Section 3.3.3), demonstrating consistent improvements across all sizes
> 3. **Cross-Architecture Validation** - Beyond Qwen2.5, we conducted experiments with Llama-3.1-8B showing strong improvements
> 4. **Data Source Diversity** - Validating across training data from three different LRMs (Gemini Flash Thinking, DeepSeek-R1, QwQ)
>
> These consistent improvements across model scales (3B-32B), architectures (Qwen, Llama), and data sources suggest our **PIR framework captures fundamental reasoning optimization principles rather than model-specific characteristics**.
>
> ### Cross-Architecture Validation: Llama-3.1-8B Results
>
> | Model | AIME24 | AMC23 | GPQA |
> | --- | --- | --- | --- |
> | **Accuracy** |  |  |  |
> | Llama-3.1-8B-LIMO-V2 (baseline) | 0.83% | 11.88% | 44.33% |
> | Llama-3.1-8B-LIMO-V2-P (PIR) | **9.13%** | **18.12%** | **45.70%** |
> | **Improvement** | **+8.3%** | **+6.24%** | **+1.37%** |
> | **Token Efficiency** |  |  |  |
> | Baseline Tokens | 19,226 | 13,875 | 11,872 |
> | PIR Tokens | 15,164 | 12,112 | 11,045 |
> | **Token Reduction** | **-21.1%** | **-12.7%** | **-7.0%** |
>
> ---
>
> ## **Q5: Benchmark Selection Criteria**
>
> > What is the formal selection criteria for the benchmarks?
> >
>
> We thank the reviewer for this question about benchmark selection methodology. Our selection follows systematic criteria ensuring fair comparison and comprehensive evaluation.
>
> **Selection Criteria**:
>
> 1. **Alignment with Prior Work** - We selected benchmarks from recent reasoning literature (LIMO, S1 papers) enabling direct comparison with established baselines
> 2. **Reasoning Diversity** - We chose benchmarks spanning difficulty levels and reasoning types—AIME/AMC for mathematical reasoning, GPQA Diamond for scientific reasoning—ensuring comprehensive evaluation across varied demands
>
> **Methodological Consistency**: Aligning with LIMO and S1 evaluation protocols ensures performance improvements stem from our PIR framework rather than methodology differences, establishing validity and enabling fair comparison.
>
> **Generalizability Validation**: Beyond established benchmarks, PIR demonstrates consistent improvements across **different data sources** (Gemini Flash Thinking, DeepSeek-R1, QwQ), **model scales** (3B-32B parameters), and **reasoning domains** including commonsense reasoning (CommonsenseQA), logical reasoning (LogiQA), and code generation (MBPP). This indicates PIR captures universal reasoning optimization principles rather than domain-specific patterns.
>
> ### Cross-Domain Performance Validation
>
> | Model | Commonsense QA | LogiQA | MBPP |
> | --- | --- | --- | --- |
> | **LIMO** | 81.33 | 77.41 | 73.2 |
> | **LIMO-P** | **82.00** | **78.04** | 72.6 |
> | **LIMO-V2** | 76.41 | 78.91 | 74.0 |
> | **LIMO-V2-P** | **80.92** | **79.28** | **75.0** |
>
> This comprehensive validation confirms that our benchmark selection provides both rigorous comparison standards and robust evidence of broad applicability across diverse reasoning scenarios.

---

> > ### Comment · Reviewer_v7KQ · 2025-08-06
> >
> > Thank you for the authors for their detailed response.
> >
> > For Q1:
> >
> > It would be beneficial if you unpack and elicit the reasoning properties of the datasets. At the moment the narrative is using the dataset as a proxy for reasoning types. Is there a more formal charaterization that you can do in terms of what each dataset represents?

---

> ### Author Response · Authors · 2025-08-06
> **Response to Follow-up Question on Formal Dataset Characterization**
>
> > Thank you for this excellent follow-up question that helps clarify our approach. To ensure we address your valuable concern accurately: are you asking for a formal characterization of the reasoning properties inherent in our training datasets rather than inferring reasoning type effectiveness through benchmark performance comparisons?
>
> If our understanding is correct, we provide systematic characterization using **Evidence 1** to directly address your question about formal dataset properties, with **three additional complementary lines of evidence** that validate this conclusion: Multi-method Validation contributes less than Verification and Error Correction to reasoning effectiveness.
>
> > Evidence 1: Training Dataset Reasoning Property Distribution
>
> Based on our systematic analysis of reasoning patterns in training data:
>
> | Dataset | Progressive Reasoning | Verification | Multi-method Validation | Error Correction |
> |---------|----------------------|--------------|------------------------|------------------|
> | LIMO (Original) | 59.7% | 11.8% | 10.9% | 17.6% |
> | LIMO-P (PIR-optimized) | **63.1%** *(+3.4%)* | **11.2%** *(-0.6%)* | **7.8%** *(-3.1%)* | **17.9%** *(+0.3%)* |
>
> **Key Insight**: PIR's quantitative importance calculation predominantly targets Multi-method Validation (-3.1% reduction) while preserving Verification (-0.6% minimal reduction) and Error Correction (+0.3% slight increase), indicating Multi-method steps contain higher functional redundancy as measured by our perplexity-based PIR metric.
>
> > Evidence 2: Model Output Reasoning Distribution Changes After Training
>
> As detailed in our Q3 response, models trained on these respective datasets exhibit corresponding reasoning generation patterns:
>
> | Model | Progressive Reasoning | Verification | Multi-method Validation | Error Correction |
> |-------|----------------------|--------------|------------------------|------------------|
> | LIMO | 28.78% | 50.57% | 20.34% | 0.31% |
> | LIMO-P | **32.83%** *(+4.05%)* | **48.95%** *(-1.62%)* | **17.89%** *(-2.45%)* | **0.33%** *(+0.02%)* |
>
> **Convergent Pattern**: Models internalize the training data distribution, with LIMO-P showing reduction in Multi-method Validation generation (-2.45%), confirming these steps are less essential for reasoning quality.
>
> > Evidence 3: Human Evaluation of Model Output Reasoning Distribution
>
> **Human Evaluation Protocol**: As reported in our Q3 response, we conducted rigorous evaluation with 4 expert annotators independently analyzing 5% randomly sampled correct reasoning traces from LIMO and LIMO-P models. Annotators classified reasoning steps into four cognitive patterns and calculated Progressive Reasoning proportions, achieving high inter-annotator agreement (κ = 0.87).
>
> **Results**: Human evaluators consistently identified significantly higher Progressive Reasoning proportions in LIMO-P traces (**68.3% ± 4.2%**) versus LIMO traces (**52.1% ± 5.8%**), representing a **16.2 percentage point increase**. This systematic validation confirms that Multi-method Validation reduction enhances reasoning concentration without compromising logical integrity.
>
> > Evidence 4: Benchmark Performance Validation
>
> Our ablation study removing specific functional reasoning types confirms this hierarchy:
>
> | Model Variant | AIME | AMC | GPQA Diamond |
> |---------------|------|-----|--------------|
> | LIMO (Full) | 56.7 | 91.9 | 67.2 |
> | LIMO (No Verification) | 59.1 | 92.5 | 70.7 |
> | **LIMO (No Multi-method)** | **60.0** | **92.8** | **72.2** |
> | LIMO (No Error Correction) | 57.5 | 92.1 | 71.7 |
>
> **Systematic Validation**: Removing Multi-method Validation achieves the highest performance improvements across all benchmarks, while removing Verification or Error Correction shows smaller or mixed effects.

---

### Comment · Area_Chair_qaHj · 2025-08-08
**Urgent reminder for reviewers**

- This is the final reminder: If you have not yet participated in discussions with authors, please do so immediately (a side note: it is a bad practice to join discussion only in the last hours of discussion period).

- Clicking “Mandatory Acknowledgement” prematurely does NOT release Reviewers from participating in discussions. Reviewers who do not participate in the discussions will be red-flagged by ACs and the system, in accordance with possible penalties of this year's responsible reviewing initiative and future reviewing invitations.

---

### Decision · Program_Chairs · 2025-09-17

**Decision:**

Accept (poster)

**Comment:**

**summary**
This paper introduces Perplexity-based Importance Refinement (PIR), a novel framework for optimizing the reasoning chains of large language models. The method distinguishes between essential "progressive" reasoning and auxiliary "functional" steps (e.g., verification, correction), and uses a perplexity-based metric to prune low-importance functional steps from training data. This results in models that generate more concise reasoning while achieving higher accuracy and efficiency.

**strengths**
- The paper proposes a principled and intuitive methodology, using perplexity to quantitatively measure the importance of each reasoning step, which is more robust than simple heuristics.
- The experimental results are strong and comprehensive, demonstrating consistent improvements in both accuracy and token efficiency across a diverse set of challenging reasoning benchmarks, model sizes, and architectures.
- The evaluation is thorough, including extensive ablation studies and cross-domain validation on tasks like commonsense reasoning, code generation, and instruction following, which confirms the broad applicability of the framework.

**weaknesses**
- The framework's initial step of segmenting and classifying reasoning types relies on a separate, powerful language model, introducing an external dependency that could affect reproducibility and performance.
- The primary pruning method relies on a fixed ratio or a simple threshold, which might not be optimal for all reasoning instances; a more adaptive strategy could potentially offer further improvements.
- The paper focuses exclusively on perplexity as the importance metric, and a deeper exploration of alternative semantic or logical-based metrics could provide a more nuanced approach to reasoning refinement.

**final descision**
This is a strong paper that should be accepted. It presents a novel, well-motivated, and empirically validated method for improving the efficiency and effectiveness of LLM reasoning, making a solid contribution to the field.